# Changes in the length of speeches in the plays of William Shakespeare and his contemporaries: A mixed models approach

**Kim Colyvas**[1], **Gabriel Egan**[2], **Hugh Craig**[3]*

1 School of Health Sciences, University of Newcastle, Newcastle, New South Wales, Australia, 2 School of Humanities and Performing Arts, De Montfort University, Leicester, United Kingdom, 3 School of Humanities, Creative Industries, and Social Sciences, University of Newcastle, Newcastle, New South Wales, Australia

* hugh.craig@newcastle.edu.au

## Abstract

Since 2007 a number of investigators have compiled statistics on the length in words of speeches in plays by William Shakespeare and his contemporaries, focusing on a change to shorter speeches around 1600. In this article we take account of several potentially confounding factors in the variation of speech lengths in these works and present a model of this variation in the period 1538–1642 through Linear Mixed Models. We confirm that the mode of speech lengths in English plays changed from nine words to four words around 1600, and that Shakespeare's plays fit this wider pattern closely. We establish for the first time: that this change is independent of authorship, dramatic genre, theatrical company, and the proportion of verse in a play's dialogue; that the chosen time span can be segmented into pre-1597 plays (with high modes), 1597–1602 plays (with mixed high and low modes), and post-1602 plays (with low modes); that some additional secondary modes are evident in speech lengths, at 16 and 24 words, suggesting that the length of a standard blank verse line (around 8 words) is an underlying unit in speech length; and that the general change to short speeches also holds true when the data is viewed through the perspective of the median and the mean. The change in speech lengths is part of a collective drift in the plays towards liveliness and verisimilitude and is evidence of a hitherto hidden constraint on the playwrights: whether or not they were aware of the fact, playwrights as a group were conforming to a structure for the distribution of speech lengths peculiar to the era they were writing in. The authors hope that the full modelling of this variation in the article will help bring this change to the attention of scholars of Shakespeare and his contemporaries.

## 1 Introduction

### 1.1 Background

In this article we explore the question of variation in speech lengths in plays by William Shakespeare and his contemporaries. Our study joins a small but growing body of work aiming to reveal hitherto hidden patterns in the plays with quantitative methods, such as the association

**Data Availability Statement:** There are two data files supporting this work. They are available from the Zenodo platform: DOI 10.5281/zenodo.6690135 and DOI 10.5281/zenodo.6690372.

**Funding:** GE received funding from the Arts and Humanities Research Council (AHRC): Grant AH/N007654/1. See https://www.ukri.org/councils/ahrc/. HC received funding from the Australian Research Council (ARC): DP160101527. See https://www.arc.gov.au/. The funders had no role in study design, data collection and analysis, decision to publish, or preparation of the manuscript.

**Competing interests:** The authors have declared that no competing interests exist.

of certain stage properties with certain genres, or the fact that styles in prose and verse diverge in plays which mix the two forms, but do not differ between all-verse and all-prose plays [1], the differences between the language of male and female dramatic characters [2], the distinctiveness of the language of tragedy [3], and an excess of instances of the determiner 'the' in the play *Macbeth* [4]. This is a separate endeavour to quantitative work in attributing plays or parts of plays to authors where their authorship is disputed, but it draws on the same foundation, the availability of literary works in digital forms and computer-assisted analysis, and on the same principles of careful validation and openness to testing by others.

Drama is defined by the fact that actors impersonate characters and speak as if interacting in real time, one speaker yielding to another. In written form we have a series of distinct passages of continuous spoken language, which we will here call "speeches," even though they may be far from speeches in the sense of extended formal orations, as in commencement speeches. Speeches in the sense adopted in this article are analogous to speaking turns in conversations in phonetics [5 p166]. A speech is conceived as the transcription of the spoken utterance of a dramatic character from the beginning of one of his or her turns to its end. Accordingly there are five speeches in the following extract from the opening of Shakespeare's play *Hamlet*:

BERNARDO Who's there?
FRANCISCO Nay, answer me. Stand and unfold yourself.
BERNARDO Long live the King!
FRANCISCO Bernardo?
BERNARDO He. [6]

The speech in this sense is one of the fundamental forms of segmentation for a play. It would be easily recognised by audiences, as well as readers, and must have been much on the minds of the playwrights who used well-established manuscript conventions to distinguish one speech from the next, most commonly drawing a short horizontal line at the end of each one. In printed plays, the first line of dialogue of each speech was typically indented by several spaces, and this whitespace aligning with the speaker's name in the left margin gave readers a strong visual indication of the work's existence as fundamentally a collection of speeches.

In this paper we focus on one simple aspect of the dramatic speeches, their length. Comparatively long speeches, such as soliloquies and speeches reporting off-stage events, declaring sentiments, or attempting to persuade, naturally capture our attention, so we might assume this sort of length is dominant. However, there is another principle at work in speech length: to sustain audience interest, regular alternation between speakers is needed. Rapid exchanges between speakers are entertaining and informative, and complement the more developed set-pieces. Then there are questions also about how other factors affect speech length. Are speeches longer in tragedy than in comedy, for instance? Is speech length something that varies consistently and markedly between authors? There are also possible changes in speech length over time, which have been the subject of a number of recent studies.

Our unit of measurement for the length of a speech is the number of words it contains. Other measures, such as the number of syllables in a speech, could be used, but while the segmentation of written text into words based on punctuation and spaces is relatively straightforward, and can be automated, segmentation into syllables is much more ambiguous and subjective. Previous studies in this field are all based on length in words. (See the fuller treatment of this aspect in the Materials and Methods section below).

We focus on plays written in England in the sixteenth century and the first half of the seventeenth century. Some hundreds of plays (many of them now lost) were performed in theatres in London and sometimes elsewhere between the beginnings of secular, English-language drama in the first half of the sixteenth century and the closing of the playhouses in 1642 at the

start of the English Civil War [7]. William Shakespeare is the best-known playwright of this era but there were many more whose plays are still performed on stage and screen, as well as read and studied, such as Christopher Marlowe, Ben Jonson, John Fletcher and Thomas Middleton [8, 9].

In this paper we aim to model speech length variation in these plays as comprehensively as possible. Previously the change in mode has been reported but not tested by reference to any potentially confounding variables and no estimates of statistical significance have been offered. In our study, date, genre, proportion of verse, play type and author were used as part of a mixed modelling approach that allowed estimation of the size of the effect of each variable and hence it: provided a ranking of the importance of each on speech length; provided formal statistical significance to test what effects were likely to be real; and provided adjusted effects sizes, so that the magnitude of effects for each variable were not biased by the other variables in the model. For example, we found that the size of the time-based changes were not influenced by differences in the proportion of the different genres among the plays in each time period. We used the mean and median as additional summary measures of the speech length distributions for plays. We also carried out extensive visualisation of the data and models to help in understanding variability in speech length. This allowed us to identify the multi-modal nature of speech length distributions and their dependence on the proportion of verse in the play and (through cusum plotting) to identify key changes in the rate and direction of the variation of the mode of speech lengths over time.

## 1.2 Previous studies

In 2005 the German scholar Hartmut Ilsemann published a study analysing the lengths of speeches in plays written by Shakespeare [10]. Ilsemann observed that in these plays the most common speech length, in statistical terms the "mode" of speech lengths, changed abruptly around 1599 from ten words to five. He suggested that what lies behind the change is that with the move of Shakespeare's company to the Globe Theatre in 1599 Shakespeare was suddenly able to control the staging of his plays. In a 2008 article he revised the numbers to 9 and 4 words respectively, since he found that he had been erroneously counting as the first word of a speech the name of the speaker added before the speech in early printed editions [11].

In 2007 MacDonald P Jackson used a table of numbers which had been published by Ilsemann online to make a new index - - counts of the number of speeches of 3–6 words as a percentage of speeches of 3–10 words - - and showed that the change in mode in Shakespeare was progressive rather than abrupt [12]. Following Ilsemann, he used the number of words as the unit of measurement for the length of a speech, while discarding speeches consisting of only one or two words and all speeches with more than ten words. Jackson used this index to help establish the chronology of some plays and parts of plays where the date is disputed. In 2017 he presented a new index based on Ilsemann's revised 2008 numbers [13]. Jackson's focus is on the association between date and speech length and he does not refer to other variables that might have an influence or other causative factors.

In 2019 Gabriel Egan confirmed the Ilsemann numbers (as of 2008) by calculating them independently [14]. In the same year Pervez Rizvi counted speech lengths in a much larger corpus, 527 plays, extending the analysis to Shakespeare's peers [15]. He confirmed the change from a mode of 9 words to a mode of 4 words in Shakespeare while showing that this is not peculiar to Shakespeare. This was evidently a pattern shared across the drama of the time.

Rizvi, following a suggestion in Jackson [12], linked the change in mode to a new propensity in Shakespeare to divide verse lines between two speakers—where part of the metrical line is spoken by the first speaker, and part spoken by a second or even a third. This change

to writing more shared lines, as Rizvi and Jackson note, was discussed much earlier in books on Shakespeare's versification by Marina Tarlinskaja [16] and George T. Wright [17], though neither investigator linked this development to any general change over time in the length of speeches. There are some discrepancies between their two counts, but they agree on the overall pattern. Lines shared between speakers were rare in Shakespeare's earliest plays, making up two per cent or so of the total, but became more common in his middle period, and more common still at the end, with the last plays. Tarlinskaja and Wright differ a little in their counts, but both have the last plays averaging over fifteen per cent, and agree that the highest percentage is in *Antony and Cleopatra*, at 18.2 per cent [16 p137-8], or 17.2 per cent [17 p294-4].

These are important findings. Even nine words seems a short speech. Almost all previous commentary had focused on longer speeches where thoughts, arguments and narratives are given fuller expression. No-one before Ilsemann had identified a collective change to yet shorter speeches over the period. It is also unexpected to find Shakespeare, who is generally assumed to be exceptional among his peers, participating in this change, following the collective pattern exactly, in the same direction, at the same time, and to the same degree.

### 1.3 The present study

Previous studies have not considered some obvious alternative explanations for an observed marked change in speech length in English Renaissance plays around the year 1600. It is possible that genre differences may play a role. For instance, we might expect the rapid repartee of comedies to favour short speeches and the declamatory rhetoric of history plays to favour long ones, and if comedies are more common after 1600, and history plays less common, then that might explain the observed change better than a simple, across-the-board drift to shorter speeches. Other factors like the proportion of metrical verse in a play, or authorial differences, or differences in the nature of the theatrical companies—for instance, boys' companies versus adult companies - - could be expected to play a role.

### 1.4 Findings

We confirm

- that the mode of speech lengths in English plays changed from nine words to four words around the turn of the seventeenth century; and

- that Shakespeare's plays fit this wider pattern closely.

  We establish for the first time

- that this change is independent of some potentially confounding factors, in particular authorship, dramatic genre, theatrical company, and the proportion of verse in a play's dialogue;

- that the time span can be segmented into pre-1597 plays (with high modes), 1597–1602 plays (with a mixture of high and low modes), and post-1602 plays (with low modes);

- that some additional secondary modes beyond 4 and 8 or 9 words are evident in speech lengths, at 16 and 24 words, suggesting that the length of a standard blank verse line (around 8 words) is an underlying unit in speech length; and

- that the general change to short speeches also holds true when the data is viewed from the perspective of the other statistical measures of central tendency, the median and the mean.

## 2 Materials and methods

### 2.1 Corpus of plays

We have 275 plays prepared from early printed and manuscript versions available for this study. We use the earliest printed version of a play, except where that version is manifestly truncated or corrupt. We avoid modern edited versions because these are available only for the more popular plays. Comparing early witnesses minimizes the effect of editorial intervention. The details of the transcriptions' provenances are in the S4 File.

### 2.2 Variables

**2.2.1 Speech length.**   Speech length is counted using the *Intelligent Archive* platform [18]. Our metric for the length of speeches is words. There would be some argument for using syllables for this purpose, since this is the basis of verse metre, and verse rather than prose is the dominant form in the plays. The difficulty here is making an accurate count of syllables as they would have been spoken. The early editions rarely follow the modern editorial practice of using punctuation to indicate omitted or additional syllables, as with "e'er" for "ever" and trisyllabic "buried." To derive the syllabification thus presented to modern readers, editors assume certain metrical norms and make complex inferences about permitted departures from these norms; computer algorithms are as yet unable to replicate this editorial judgement. Thus we cannot yet reliably automate the counting of syllables in early printed editions of plays. In counting words we align ourselves with earlier studies, which have all used words as the measure of length, based on a relatively unambiguous segmentation that can reliably be detected from the punctuation and spacing in digital transcriptions of early editions.

We say "relatively unambiguous" because in the play texts we use for counting contractions are expanded, so that Horatio's line "Not when I saw't" becomes "Not when I saw it" (*Hamlet* [6] 1.2.456). This means a variation in the number of words arrived at compared with counting contracted forms as single words. For some, but not all of the plays in our corpus, the XML mark-up also preserves the original version with contractions. We took ten randomly selected examples of these play texts and compared the word counts for the contracted and expanded versions. There were 7,125 speeches in all. 5,316 speeches are the same length in words in both versions. Overall there is a 1.9% difference in the two columns of speech lengths, based on dividing the difference in words by the average of the two counts.

**2.2.2 Other variables.**   We start with the date variable, which underpins our main hypothesis that speech lengths changed over time in the works studied. Surviving evidence of theatrical business practices - - most importantly the records of impresario Philip Henslowe - - show that professional plays were usually performed within weeks of being written, while the dates on the title pages of printed plays show that publication in book form typically followed some years later. Farmer and Lesser's *DEEP Database of Early English Playbooks* [19] conveniently provides well-sourced dates of first performance (sometimes a best guess) and of publication for all printed plays of the period. For consideration of dramatists' changing writing styles we focus on the date of first performance, since it is closest to composition, as do previous studies discussed above. For first-performance dates we use *DEEP*'s data for printed plays, supplemented by other sources (typically modern scholarly critical editions) for manuscript plays.

We include four other variables in the study as potentially associated with speech length, and as possible confounds with date: author, genre, play type, and the proportion of verse lines to all lines in a play.

Authorship's place as the pre-eminent determining factor in literary style has been challenged in recent decades, but quantitative studies have shown that it has a demonstrable role

in the differences between literary works [20]. We rely on the DEEP Database [19] for authorial assignations of plays. Our metadata sources assign an author to most plays and categorise the rest as of uncertain authorship. We categorise each authorial team, whether single or multiple, as a separate author, group authors or authorial teams each with a single play together as a group, and similarly group plays of uncertain authorship together as a single category.

Dramatic genre is recognised by modern scholars as a primary influence on the styles of plays. Early printed editions highlighted genre, as for example in the ordering of the plays in the Shakespeare First Folio of 1623 into comedies, tragedies, and histories [21, p166]. We put the genres in the *DEEP Database* into five groups: in the first, classical legend (comedy), comedy, domestic comedy and romantic comedy, which we refer to in the paper as "comedy"; in the second, tragedy (only), referred to as "tragedy"; in the third, allegorical history, biblical history, classical history, foreign history and history, referred to as "history"; in the fourth, tragicomedy (only), referred to as "tragicomedy"; and in the fifth, burlesque romance, classical legend, classical legend (pastoral), classical myth, domestic drama, heroical romance, moral, pastoral and romance, referred to as "other." See the S4 File for play by play assignations.

Theatrical companies featuring only boy actors introduced a new satirical, learned and self-consciously theatrical style of play in the period 1599–1613 [22]. We include a variable for play type, referring to the category of theatrical company the play was first associated with, being boys' professional companies, adult professional companies, and a third, miscellaneous category of amateur companies and plays not associated with any company, plays written to be read and not performed. We rely on the *DEEP Database* [19] for assignations.

The distinction between play dialogue in verse and in prose is, for the most part, obvious in the early printed texts by its representation in typography - - a line-ending being marked by whitespace and a beginning by capitalization of the first letter - - and was often commented on by the playwrights, and figures largely in modern discussions of the plays [23]. (Where early printed books have lines of verse incorrectly typeset as if they were prose, and vice versa, modern scholars can almost always nonetheless distinguish the two forms using their knowledge of the stress patterns in early modern English, and modern editions and studies make the necessary corrections.) There is reason to believe, therefore that the differing proportions of verse and prose in a play might have an influence on speech length as on other aspects. Our texts are not marked up to show which passages are in verse and which are in prose. The *Catalogue of British drama* volumes [24], however, provide figures for the proportions of verse and prose in a given play. We can use these numbers to group plays by their proportions of verse and prose.

## 2.3 Data files

Two data files were used for analysis, the larger with 212,547 individual speeches (identified by speaker) from 275 plays was computationally analyzed to produce for each play its mode, median, and mean speech length. These three statistics for each play were then added to the other metadata previously mentioned (date, genre, proportion of verse lines to all lines, play type and author) to produce a new summary dataset to be subsequently investigated for correlations between the metadata it holds. It is common in previous studies of speech length in these plays to use the mode as the summary measure (Jackson [12] is an exception mentioned in the Previous Studies section above) and we follow this practice. This is intuitively the obvious way to deal with the particular distribution of speech lengths, which has a long tail at the high end where the occasional speech is some hundreds of words long. A calculation of the mean would be skewed by such rare outliers to which the mode is relatively immune. We nonetheless present some results with the mean and the median of speech lengths to take advantage of the different perspectives they offer on speech length variation.

## 2.4 Statistical methods

We used the R statistical software environment [25] for analysis. A detailed report containing all results and commentary is provided in the S1 (the main analysis) and S2 (a subsidiary analysis comparing normal and Poisson distribution models) Files. In the S3 File we provide an examination of the distribution of 15 plays in which 2 or more modes were found and a discussion of the approach to decide on a single mode. We also include the data files (S4 File individual speeches and S5 File summaries by play) and the R markdown scripts (S6–S8 Files) needed to reproduce the three detailed reports in S1–S3 Files. Some results and explanations in the reports are referred to but not provided in the main text. We fit Linear Mixed Models (LMMs) on the play version of the data set using the lmer program with maximum likelihood estimation in the lme4 package [26] and the lmerTest package [27] to provide significance tests for the categorical variables. The outcomes assessed were speech length (as mode, median or mean) and verse-prose proportion. In each case, authorship was modelled as a random effect with the variables time period, genre, play type and verse-prose proportion (i.e. also being used as an explanatory variable) being modelled as fixed effects. Time period was a grouping variable derived from date of first performance using a cumulative sum (cusum) charting approach to identify step changes in the mode. Verse-prose proportion was categorised into bands to provide an alternative way of examining relationships associated with it. Diagnostic checks of the models were based on assessing the distribution of residuals and the distribution of the random effects. The emmeans package [28] was used for plots of Estimated Marginal Means (EMMs) and significance of differences between means. Statistical significance was set at the .05 probability threshold.

At an editor's suggestion additional modelling for the mode was done using the Poisson distribution for the error term via a Generalised Linear Mixed Model (GLMM) using the glmer program as an alternative to the normal distribution used in LMMs. The goodness of fit of GLMM Poisson models (using identity or log links) was compared with the LMM models using the Akaike Information Criterion (AIC) of 10 or more lower to indicate a substantially better fitting model [29]. The process led to the identification of 2 influential plays that were very different statistically to the remainder and these were removed from all analysis so that the results reported below were based on 273 plays. This analysis also identified that the residual variability was not constant, so a relationship was developed for the residual standard deviation (SD) as a linear function of the absolute value of the residuals against predicted value [30]. This was used to create observation weights ($w = 1/SD^2$) that were applied in the LMM fitting process. Pearson residuals (residual/SD) were used to assess the effectiveness of the SD function used in the weighting. The result of the comparison was to confirm the normal distribution based LMM was a better fit for the mode and this was the model chosen for the mode analyses (as described above). The two influential plays that were removed were *The tragedy of Mariam* by Elizabeth Cary with an extreme outlier mode of 33 and *Philotas* by Samuel Daniel with a Pearson residual of 9. These plays were also excluded for the mean and median analyses. The process that led to these exclusions is documented in S2 File. There is literary support for excluding Elizabeth Cary's play. Cary was a highly educated woman who moved in a narrow circle of aristocratic patrons of the arts, with no personal connections that we know of to professional, or even amateur, drama. Her play's intended consumption seems to have been not in public performance but as a reading text for her circle, even if scholars have noted that it has some theatrical qualities that set it apart from the run of so-called closet drama [31, 32]. Hence it is not surprising that it did not show the trends found in plays written by professional dramatists for performance on the London stage. The anomalous distribution of speeches by length in *Philotas* may be explained by the fact that it (like *The tragedy of*

*Mariam*) is an imitation of the plays of the Roman tragedian Seneca, which are characterised by long speeches of rhetorical persuasion and description [33].

### 2.5 Linear mixed model

A mixed model contains two kinds of effects: fixed and random. The variables modelled as fixed effects were listed above and chosen because of the interest in the specific values for different levels of those variables. Author, on the other hand, was modelled as a random effect as there was no specific interest in distinguishing between authors in this sample. Rather the interpretation of author variability will be wider, having application to all authors, not just those included in the study.

The details behind the mixed model equations can be found in [26]. A simplified representation is provided below based on the variables examined in this study.

$$y_m = \mu + g_i + p_j + t_k + \beta v + a_l + e_m \tag{1}$$

where $y_m$ is the data for each of the m plays in the sample, $\mu$ the overall mean of the data with the first 4 terms the fixed effects, $g_i$, for i = 1,..,5 being the 5 fixed effects for genre, and similarly for $p_j$ the 2 play type effects, $t_k$ the 3 time period effects and $\beta$ the regression coefficient for v a continuous variable for proportion of verse in a play. The $a_l$ are the coefficients for the author random effect and finally $e_m$ the residual or unexplained variation for each play. The last two terms in Eq 1, the author random effects $a_l$ and $e_m$, are summarised as standard deviations, $\sigma_a$ for author variability and $\sigma$ for residual variability.

As residual variation was found to vary as a function of model predicted values a weighted analysis was carried out so that observations with higher variability would be down-weighted relative to those with lesser variability. In [30] various methods were surveyed for determining a function to explain how the standard deviation (SD) of residuals would vary with an explanatory variable. The method used in this study was to regress the absolute value of residuals on predicted value. The linear regression line describing the standard deviation of residuals ($\sigma_p$) as a function of predicted value was used to determine the weights used in the model, calculated as $w = 1/\sigma^2$.

## 3 Results

### 3.1 Mode of speech length and the time variable

Plotting the modes against year of first performance in our sample confirms the marked dip around the turn of the seventeenth century observed in previous studies (Fig 1).

To establish boundaries between the early period and what appeared to be a transitional period, and the late period, we plotted the mode observations in a cumulative sum (cusum) analysis. A cusum chart is used to interpret a process in terms of steps and helps in identifying the time at which step changes occurred. Cusum charts are usually plotted with time data that is obtained at regular intervals. The method has been adapted here to monitor the process of play production which is very irregular in terms of time. This does not invalidate the method, however, so long as the underlying process is operating at a steady level till a change occurs to a new consistent level after the change point. The plays were ordered by year and multiple plays written within a year can be considered as random samples within that year as they were ordered alphabetically based on play title. Therefore, the best that could be achieved by the chart with the play data would be to identify changes to the nearest year.

Fig 2 shows the results.

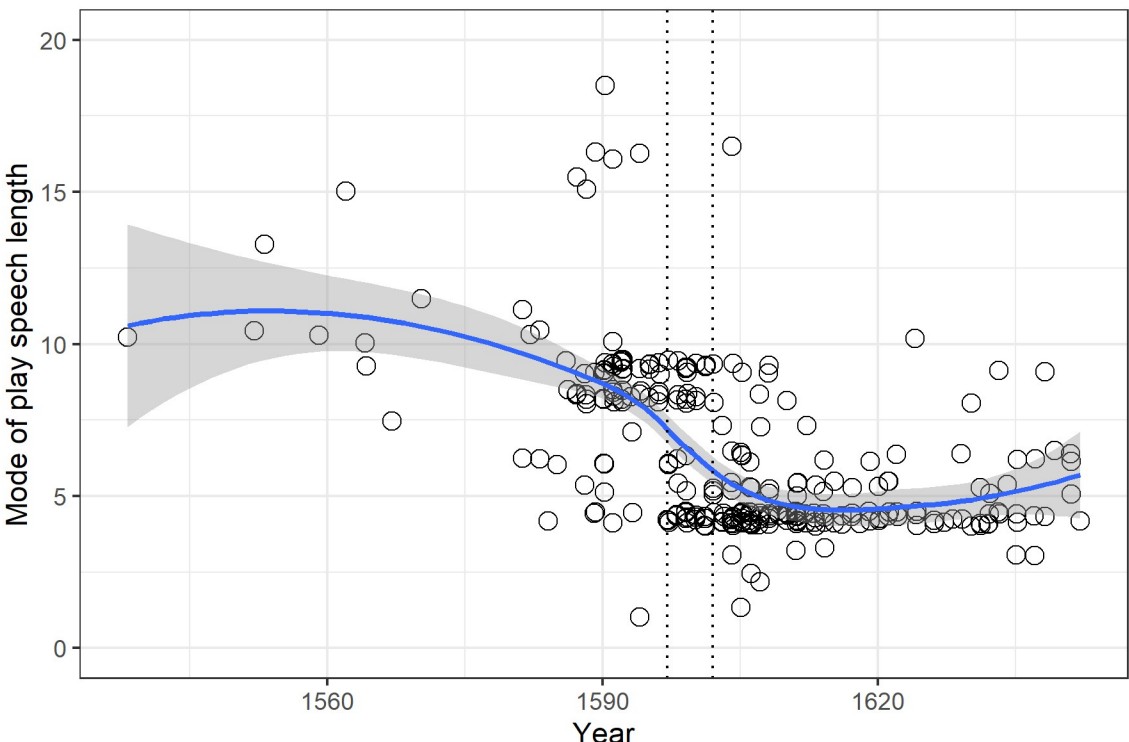

**Fig 1. (S1 Fig 1 in S1 File), Plot of mode for all plays with the solid line formed by a loess smoother showing the trend to lower values over time.** Two dotted vertical lines border the period 1597 to 1602. The mode was jittered.

The first main segment of the cusum plot, observations 1 to 80 (ending at the second vertical dotted line) had a positive slope indicating the mode was at a level (step) above the arbitrarily chosen target value of 6. As the slope of the line was generally linear this indicated that the mode was stable. Within this first segment, however, two sub-sections can be identified. Observations 1 to 14 (years 1538 to 1583) appear to have a steeper slope so if this was treated as the first step the mode would be on average 9.9. For the second section of observations 15 to 80, the slope was less steep than the first section indicating a lower mode, mean of 8.5. However, this more fine-grained division is less consequential than the change to the second segment, where the slope is approximately horizontal, indicating the mean was approximately the same as the target level for the plot. This section covers observations 81–126 (years 1597 to 1602), with the mean of the modes being 6.1. Then there is a second large change point, at the beginning the of the final section, which covers observations 127–273 (years 1603 to 1642). In this section the mode was at its lowest level, with a mean value of 4.7.

For a quantitatively based segmentation of the time span, for later analysis, using these cusum results, we divided the overall time span into three sections, 1538 to 1596, 1597 to 1602, and 1603 to 1642. Table 1 shows how many (n) of the 273 plays fall into each period.

## 3.2 Mode of speech length, time segmented into three periods, and the verse proportion variable

**3.2.1 Three groups for proportion of verse lines.** As discussed in the Materials and Methods section above, we have for most of the plays a figure for the proportion of verse lines for the play, the remainder being prose lines. We divided the plays into four groups depending

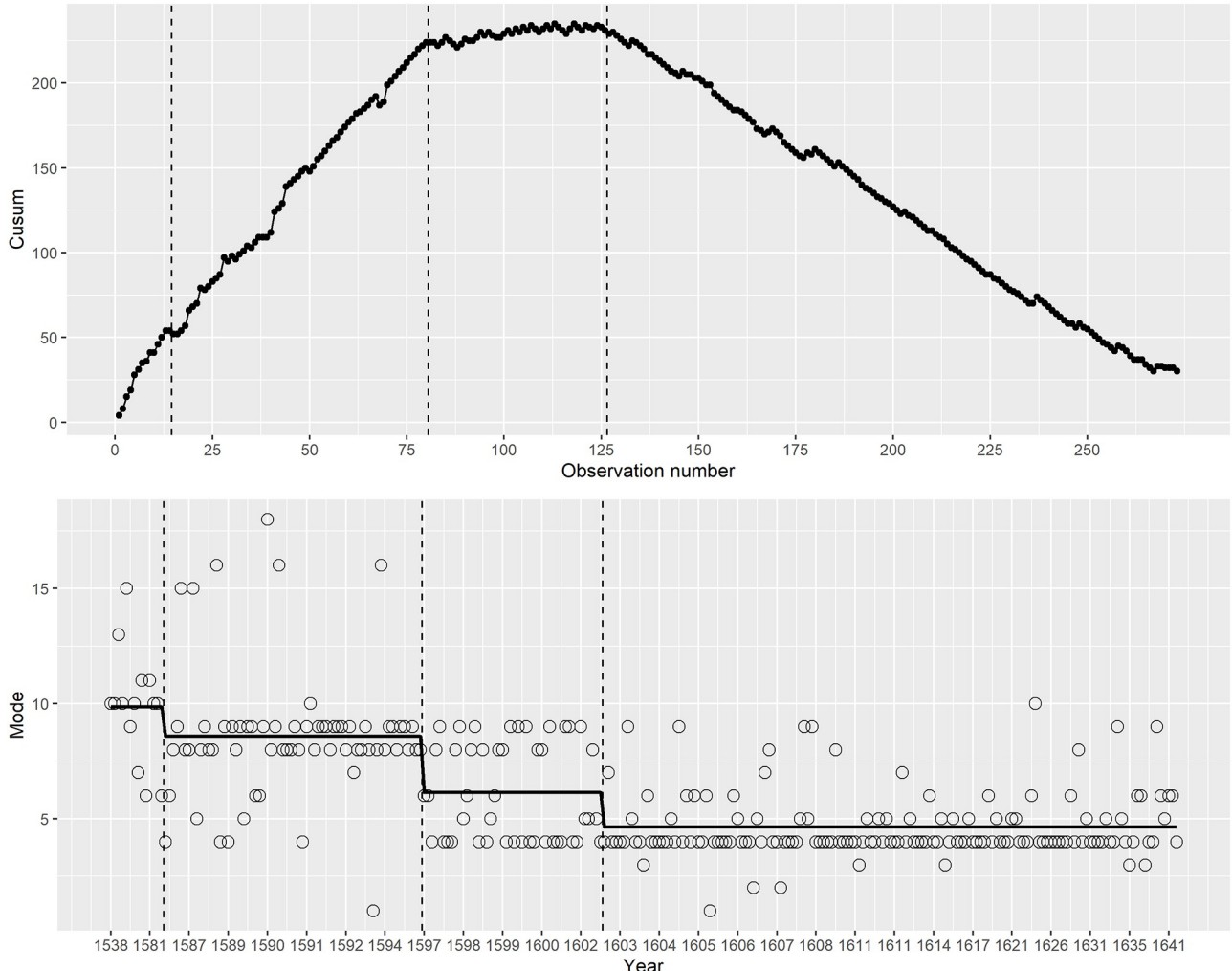

**Fig 2. (S1 Fig 2 in S1 File) The upper plot is the cusum plot of mode for each play.** The approximately straight-line segments with different slopes indicate the time periods that are interpreted as steps. Time change points are where the line segments change slope. The lower plot is the mode data with the solid line showing the step function interpretation for the mean of the mode using the change points in the cusum plot.

on their proportion of verse lines. The divisions were a compromise between keeping approximately similar numbers of plays in each group and maintaining a good spread of proportions within each of the categories. As can be seen in Table 2, most of the 273 plays fall into a narrow range at the higher proportion end of the scale compared to fewer plays spread over wider range of proportions at the lower end of the scale, and a few plays for which this data was missing.

**Table 1. Plays segmented into three periods.**

| Span of years | n (Number of plays from this time-span) | Percentage of the 273-play dataset falling into this category |
| --- | --- | --- |
| 1538–1596 | 80 | 29.3 |
| 1597–1602 | 46 | 16.8 |
| 1603–42 | 147 | 53.8 |

**Table 2. Plays segmented into three categories of proportions of verse lines.**

| Proportion of verse lines in the play | n (Number of plays having this proportion) | Percentage of the 273-play dataset falling into this category |
|---|---|---|
| 0–0.30 | 37 | 13.6 |
| 0.30–0.70 | 56 | 20.5 |
| 0.70–0.90 | 76 | 27.8 |
| 0.90–1.00 | 94 | 34.4 |
| Missing values | 10 | 3.7 |

**3.2.2 Histograms of modes in time period and verse proportion panels.** Segmenting the date of first performance of the plays into three periods and the verse proportions into four groupings makes it possible to visualize the interaction between these two variables. In Fig 3 we used the larger data set containing the individual speeches to produce histograms of the modes of speech lengths 1 to 30, divided into twelve panels. The four panels within each row of the chart are the four groupings of proportions of verse lines and the three panels within each column are the three time period groups. Thus each panel represents all the plays falling into a particular category for proportion of verse lines within a particular period.

Within each panel we show by column height the percentage of speeches with one word, to the extreme left, then the percentage of speeches with two words, up to thirty words on the extreme right. The tallest of these columns is the mode, the value for speech-length found more often than any other value.

Overall it is clear the speech length distributions vary with both the variables. We first discuss the changes between the time periods, moving down the columns of the chart. In the top

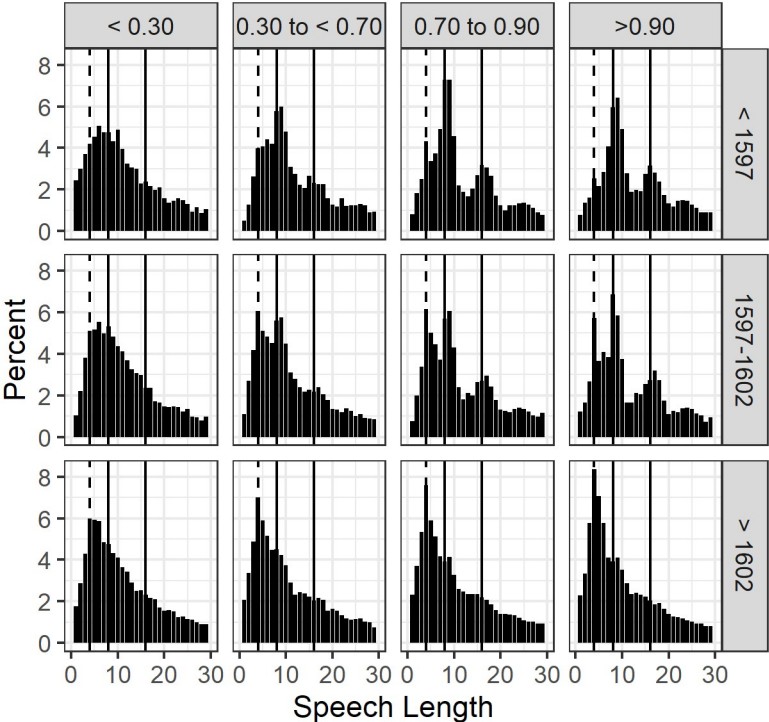

**Fig 3. (S1 Fig 4 in S1 File) Distribution of lengths of speeches (the range being from a mode of 1 word to a mode of 30 words) by verse and time period groups.** The vertical markers are at speech lengths 4, 8 and 16.

left component chart, plays with <0.30 verse lines before 1597, the highest column is the sixth from the left, speeches with six words. In the middle chart in this column (plays with <0.30 verse lines between 1597 and 1602) speeches with six words still have the highest percentage. Then in the bottom left chart (plays with <0.30 verse lines post-1602) the fourth column is the highest. The speech length mode has moved from six words to four words.

Moving to the far right column, speeches from plays with more than 0.90 verse lines, in the top chart (plays before 1597) the mode is 9 words, with a second peak at 16 words. Then in the middle chart (plays from 1597 to 1602) the mode is 8 words, with a second peak at 4 words, and a third peak at 17 words. In the bottom chart (plays after 1602) the mode is 4 words.

We now discuss the chart from the viewpoint of changing proportions of verse lines in plays. Looking along the first row of charts for the period before 1597, when the proportion of verse in a play is below 0.30 (top left panel) the distribution is a relatively smooth skewed shape. However, in the next 3 charts, with increasing proportions of verse lines within a play (0.30 to just below 0.70, 0.70 to 0.90, and more than 0.90) the single distribution appears to change into a mixture of distributions. An obvious and dominant distribution appears with a mode in the region of speech lengths 8 to 9. Interestingly a second but smaller distribution becomes obvious with a mode at 16 for the two highest verse groups as well as a third even smaller distribution with mode at 24. The separation of the distributions into mixtures with different modes as the proportion of verse in a play increases supports the notion these modes are related to the number of words in a line of verse. The dominance of the mode at 8 or 9 suggests a predominance of speeches of one metrically complete line of verse, with the other two peaks at 16 and 24 being related to two or three metrically complete lines of verse. That is, the preference for metrical completeness entails a preference for speech lengths that are whole multiples of about 8, which is the typical number of words in a line of the most common meter, iambic pentameter.

The pattern of modes changes dramatically in the bottom row of the chart, covering the period after 1602, where all three of the higher verse groups distributions have a single dominant peak with only a slight hint of the peaks at 16 and 24. The lowest verse group, < 0.30, which did not have a strong multi- peak distribution in the pre-1597 graph, shows a shift of its mode from 6 in the pre-1597 period to a lower value of 4.

The middle row of plots for the period 1597 to 1602 contains a mixture of the distributions of the earlier and later periods. The appearance of modes at 4 for the 3 highest verse groups, while other peaks at 8 to 9, 17 and 24 are still evident, is indicative of the transition between the pre-1597 and post-1602 distributions. This can be seen in Fig 2 where the mode is plotted over time. Clearly there is a mixture of modes in the transitional period with some plays having modes of about 8 and quite a number having modes of 4. After 1602 the predominance of plays with modes of 4 is apparent.

The multi-peak nature of the speech lengths distributions in the early and transitional periods is a finding new to this study. The most common verse type of the period is the iambic pentameter with ten syllables, though shorter lines and lines with extra syllables are permitted variations around this norm. A ten-syllable line may have ten words, if each has one syllable, but it would be more common to have one or more multi-syllable word and therefore nine, eight, or even seven words. Peaks of 4, 8, 16 and 24 words in Fig 3 suggest a link between the mode and half, one, two or three multiples of the pentameter line, suggesting that this unit figured in the compositional habits of the writers, and has an effect, although weaker, even in plays composed entirely or mostly in prose.

In summary the change from a multi-peak distribution in the period prior to 1597 to a primarily single-peaked distribution after 1602 points to a substantial change in the style of the

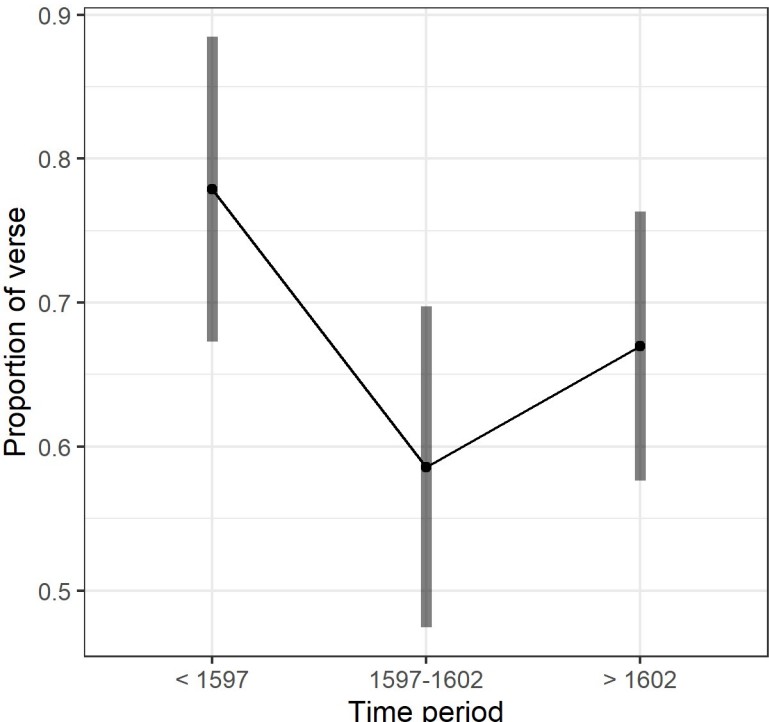

**Fig 4. (S1 Fig 5 in S1 File) Estimated marginal means and 95% Confidence Intervals of the proportion of verse in plays for the three time periods.**

writing that began about 1597 and was largely complete by 1602, irrespective of the proportion of verse lines in a play.

Given that the proportion of verse lines in a play is clearly a factor in the distribution of the modes of speech length, we explored the relationship between verse line proportion and mode of speech length more directly.

**3.2.3 Verse proportions in the three time periods.** We ran a Linear Mixed Model (LMM) analysis with verse proportion as the outcome variable to see how it varied between the time groups. The time period effect was significant, $F(2,254.1) = 7.2$, $p = .001$. Examination of the plot of the Estimated Marginal Means (EMM) in Fig 4 shows that pre-1597 plays had significantly higher proportions of verse than later plays in 1597–1602 period for the post-1602 period ($p = < .001$ and .02 respectively). The mean proportion of verse in plays was about 78% before 1597, 59% in the transition period 1597–1602 and 67% after 1602. The post-1602 period was not quite different to 1597–1602 ($p = .06$).

Was this change in verse proportion over time the underlying reason for the speech length changes observed in the mode and for median and mean? This question was examined by some additional data exploration in the following sections.

**3.2.4 Chart of modes with proportions of verse in three time periods.** In Fig 5, below, we plotted the proportion of verse within plays against the mode, for the three time periods. Overall there did not appear to be a consistent relationship between the proportion of verse and mode within each time period with the Generalised Additive Model (GAM) based smoothing lines within time periods being both close to linear and not varying much about the horizontal. This was investigated further statistically with LMMs treating proportion of verse as a linear predictor. The detailed results and discussion are available in the S1 File.

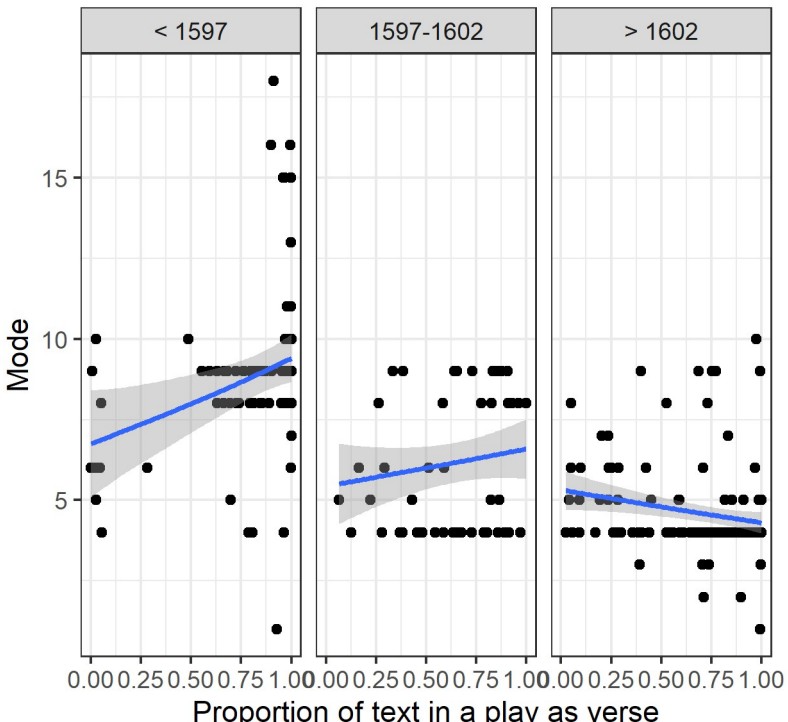

**Fig 5. (S1 Fig 6 in S1 File) Mode of speech lengths plotted against proportion of verse in a play.** A GAM smoothing line and 95% Confidence Interval was added within each time period.

Overall the conclusion was that verse proportion did not appear to be important for the mode. This was confirmed (see below) by modelling with additional explanatory variables.

**3.2.5 Verse proportion by time and genre.**  Additional graphical exploration of the verse proportion over time (see the two part Fig 6, below) reveals that quite a few plays in the period 1597 to 1612 (the transition period 1597 to 1602 and the following decade, as shown by dotted vertical lines) had verse proportions below 50% and that these were predominantly comedies. Comedies, and miscellaneous-genre plays, range from 0% verse—entirely prose—to 100% verse. After 1612, however, only a few comedies had verse proportions less than 50%.

**3.2.6 Verse proportion for comedies only versus speech length.**  Comedy is the most common genre in the sample and shows the greatest range of verse proportion (Fig 6(b)). If there was a relationship between verse proportion and the distribution of speech length, we would expect it to appear within the comedy genre. In Fig 7 we plotted the mode, median and mean of speech length in comedies against verse proportion to test this.

The fitted GAM smooth lines in the plots suggest there was no relationship. We conclude that the time-based changes seen in the analysis were not related to differences in the ratio of verse to prose. It is the more fundamental changes in the form of writing that made for shorter speeches. There is a relationship between the peakiness of the mode of speech length and the verse in a play (Fig 3), suggesting that writers respected an implicit standard of a half, one, two or three verse lines in the most common lengths of speech they wrote, but this is independent of the underlying tendency of writers to write shorter speeches in the transition period of 1597 to 1602 and then to continue to write speeches of the same (shorter) length for the rest of the period studied. This leaves date of first performance as the top-level predictor of the mode of speech length.

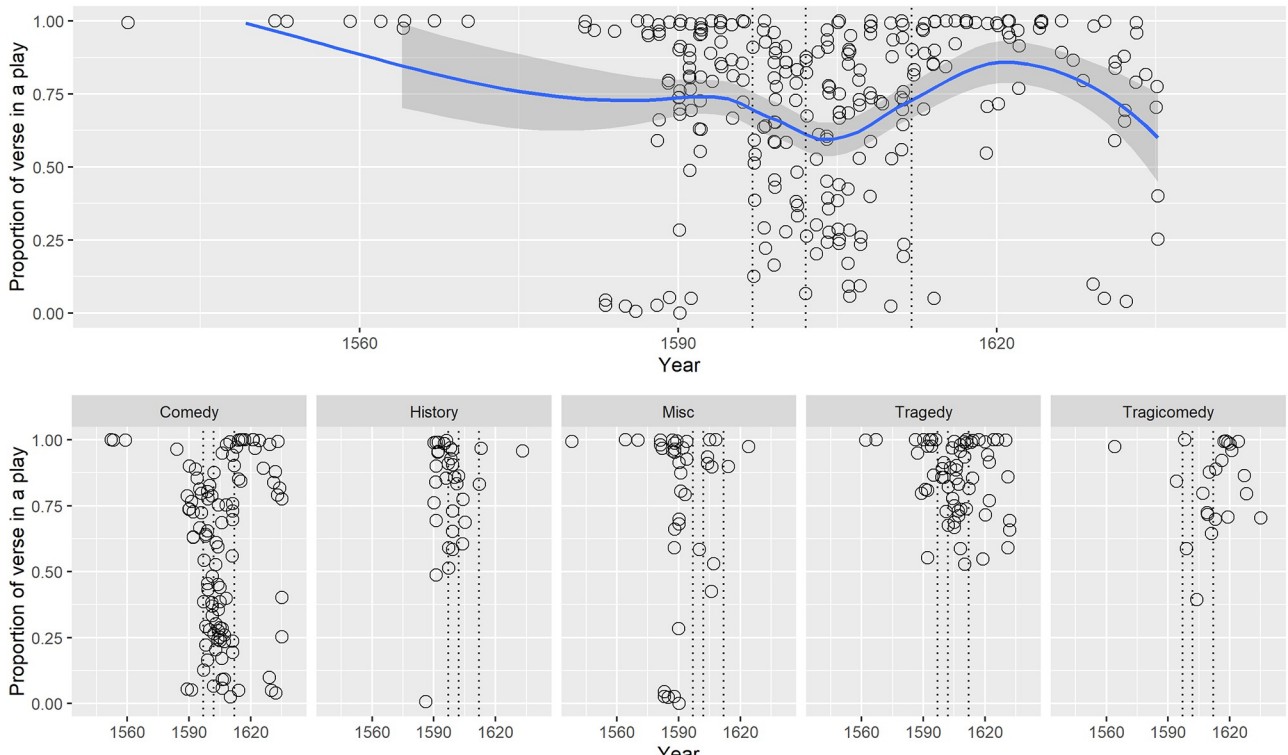

**Fig 6. (S1 Fig 7 in S1 File) (a) Proportion of verse in a play by date (upper)and (b) proportion of verse in play by date, in five genre groups (lower).**

## 3.3 Mode and time period, genre, and play type

Having looked closely at the proportion of verse as a factor underlying the association of the mode of speech length with time period, and having found no significant relationship, we turn to two other factors, genre and play type.

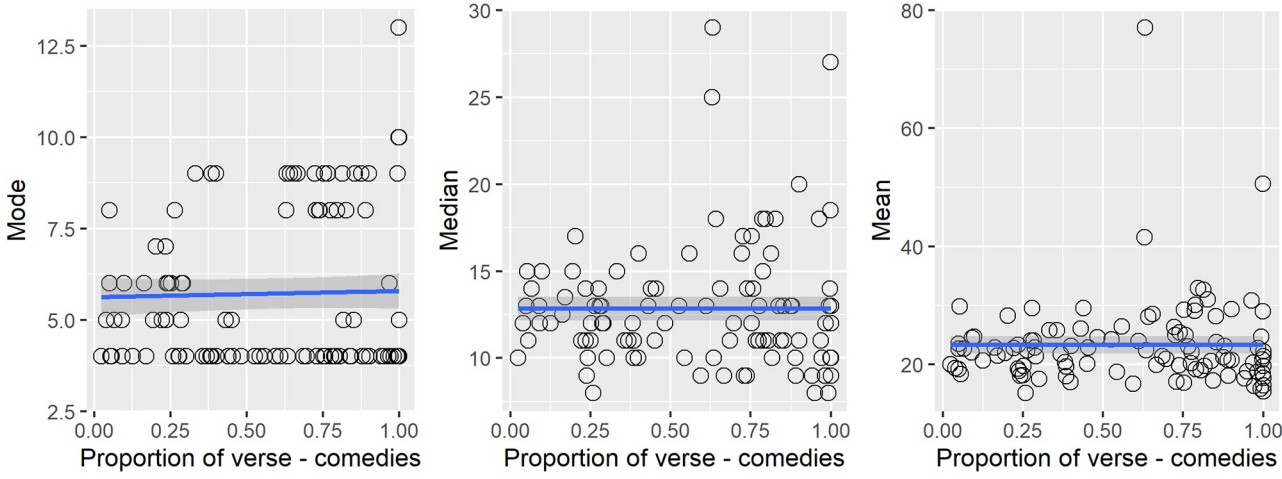

**Fig 7. (S1 Fig 8 in S1 File) Comedies only: Mode, median and mean of speech lengths versus proportion of verse.** The fitted line is based on a GAM smoother.

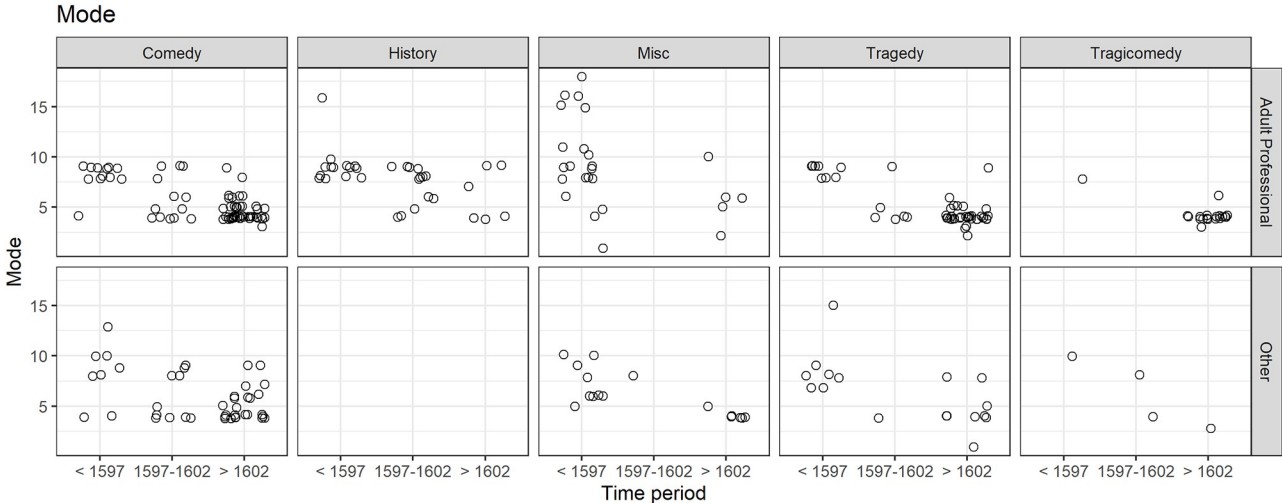

**Fig 8.** **(S1 Fig 9 in S1 File) Mode of speech length (with jittering) by genre, play type and time groups.**

By way of introduction, Fig 8 plots all the mode data by time period (see the three-way division in the horizontal axis in each of the component charts), genre (the five vertical pairs of charts) and play type (the two rows of charts).

We note the absence of history plays in the play type category Other (comprising 59 Boys' Professional and 20 Miscellaneous plays). That is, all the history plays were performed by adult professional companies. Tragicomedies were performed mostly after 1602, by adult professional companies, with lower modes. The lower modes after 1602 are evident in comedy and in tragedy in both adult professional and Other play types.

Using a linear mixed model (LMM) with author as a random effect and model weights based on an SD function for non-constant variance in the model residuals (comprehensively examined and explained in S2 File), we modelled the variation in the mode to see if any of genre, play type and time period (all as main effects) were significant variables. Genre and time period were significant, genre $F(4,268.8) = 4.52$, $p = .002$, play type $F(1,141.0) = 0.11$, $p = .74$ and time period $F(2,260.5) = 55.8$, $p < .001$. Tests of model assumptions were carried out and found to be satisfactory (see S1 Fig 11 and S1 Fig 12 in S1 File [identical to Fig 10, below] and associated explanatory text).

From this model, estimated marginal means (EMMs) were calculated for each of the three main effect variables, as shown in Fig 9. It can be seen that the differences between the EMMs for different genres or play types are relatively small compared to the differences between the time periods, being the dominant effect. Letters on the plot indicate categories that are different when the letters differ and not different when the letters are the same. For example, each time period was statistically significantly lower than the previous, hence the a, b, c progression in letters. For genre due to the complexity of the pattern of significance among the pairwise differences a significance level of .11 was chosen rather than .05 to make interpretation of the pattern simpler. Historical plays had the highest mode at about 7.5, being higher than the second tier group of comedy and miscellaneous at about 6.4 (b) with tragedy and tragicomedy being similar at the lowest level of about 5.8 (c).

### 3.4 Speech length by author

In the model above author was modelled as a random effect so the variation in author means was estimated using a normal distribution (with $\sigma_a = 0.36$). The assumption of normality of

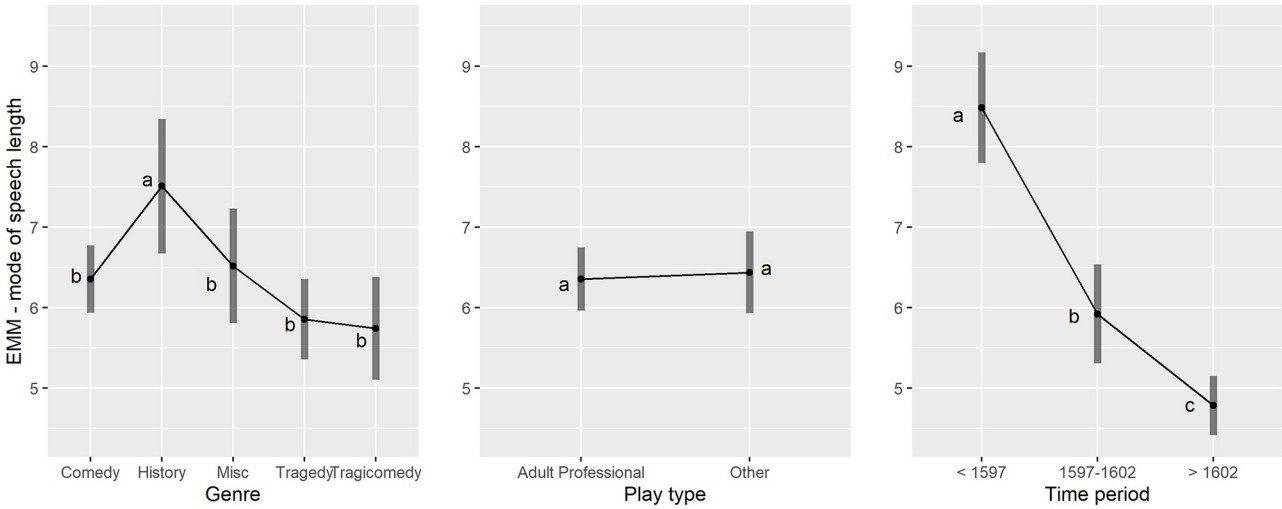

**Fig 9. (S1 Fig 10 in S1 File) Mode of speech length: EMMs and 95% Confidence Intervals from the final LMM model with main effects for the three explanatory variables of genre, play type and time period.** Within each plot letters have been added to show which levels are statistically different, categories with different letters are different, letters the same not different.

this random effect was well met (the two right hand graphs in Fig 10). The caterpillar plot at the left hand side of Fig 10 on the *x* axis shows the mean (Best Linear Unbiased Prediction, BLUP, the $a_l$ in Eq 1) for each author along with their 95% Confidence Intervals. These indicate the mean value (of the mode) for each author after adjusting for the fixed effects in the model. The differences between them can be used to assess author differences in speech lengths. They range from the lowest level of about -0.3 for the author with the lowest mode (John Marston) to the highest of about +0.4 for Richard Broome and the Uncertain authors

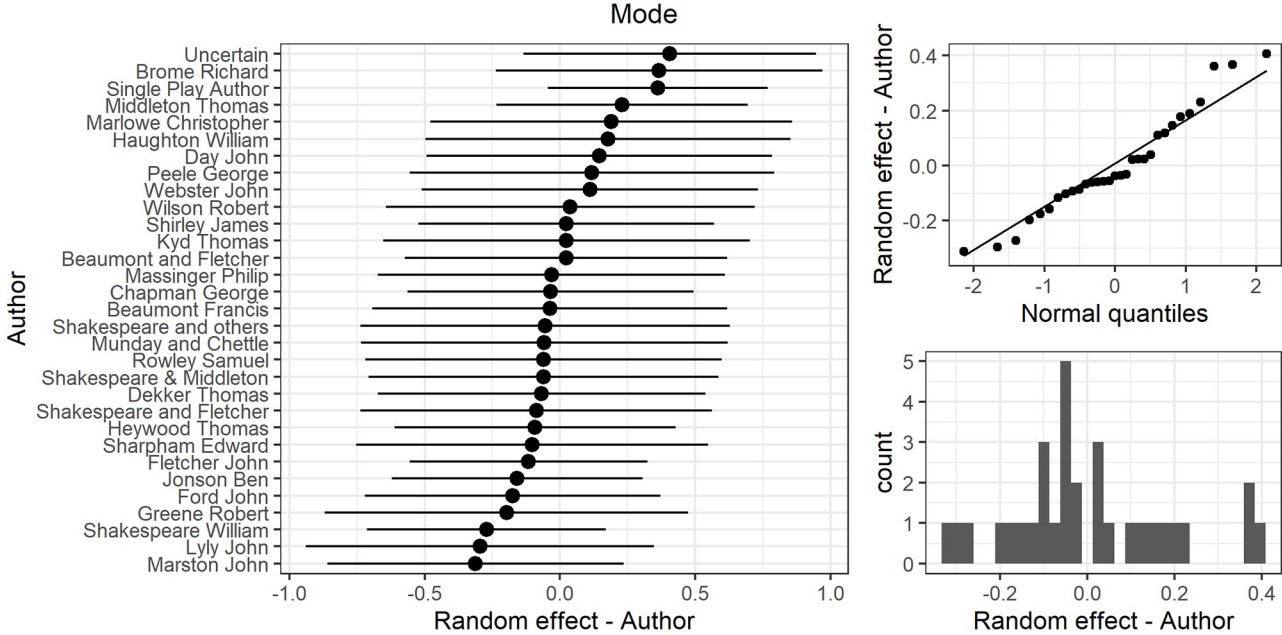

**Fig 10. (S1 Fig 12 in S1 File) Distribution of author random effects for the mode.**

category. As an example to help understand the author random effects, consider the EMM for plays after 1602. Without consideration of authorship, that is if we average over all authors, the EMM would be 4.8. If, however, predictions were to be made for the mean of the mode for specific authors in this period, then for John Marston it would be 4.8–0.3 = 4.5 and for Richard Broome 4.8 + 0.4 = 5.2.

However, the choice of author as a random effect rather than fixed carries with it a wider interpretation. The individual author effects above were estimated using the BLUPs, however beyond this the set of authors is considered as belonging to a distribution of author effect. The standard deviation of 0.36 estimated for the author distribution coupled with the normal distribution as a model can be used to predict the likely range of effect for all authors, not just those included in the study. The estimated author effects (the BLUPs in Fig 10) would be considered to be part of a random effect normal distribution with mean zero and $\sigma_a$ = 0.36. Using the properties of the normal distribution we could say 95% of authors would be expected to have effects on the mode ranging ±1.96 x 0.36 = ±0.71, range 1.42, or 99.7% of authors would be expected to have a range of 3 x 0.36 = ±1.08. As the check of model assumptions for the random effect coefficients in Fig 10 showed good agreement with the normal distribution the model predictions for the range of all author effects should have good reliability.

Comparing the size of the author differences with a range of 0.7 (-0.3 to +0.4) with the decrease over time with a range of 3.7 (EMMs 8.5 down to 4.8) shows that the impact of the event that led to a general decrease in mode for plays before and after 1600 was much larger than individual author differences.

1597 to 1602 was the transition period. We were interested to determine whether some authors might have been pioneers in transitioning their writing to lower modes (4, 5 or 6).

Table 3 shows the number of modes at each value from 4 to 9 for the thirteen authors and authorial teams with plays in this period, along with the "Uncertain" category (those plays where the author was unknown).

Of the eight authors with more than one play in this period in our sample, five have plays both at the low end (modes of 4, 5, or 6) and at the high end (7, 8, or 9), so for them the pattern is mixed. Of the three remaining authors with more than one play, two, William Haughton and Thomas Heywood, have plays exclusively with modes of 7, 8 or 9, and one, Ben Jonson, has plays—in his case five of them—exclusively at the low end, with modes of 4, 5 or 6. Jonson

**Table 3. Number of plays with modes ranging from 4 to 9 for each author or author group in the transition period 1597 to 1602.**

|  | 4 | 5 | 6 | 7 | 8 | 9 |
|---|---|---|---|---|---|---|
| Brandon, Samuel |  |  |  |  | 1 |  |
| Chapman, George | 1 | 1 |  |  | 1 | 2 |
| Chettle, Henry |  | 1 |  |  |  |  |
| Dekker and Webster |  | 1 |  |  |  |  |
| Dekker, Thomas | 1 |  |  |  |  | 2 |
| Haughton, William |  |  |  |  | 1 | 1 |
| Heywood, Thomas |  |  |  |  | 1 | 1 |
| Jonson, Ben | 3 | 1 | 1 |  |  |  |
| Marston, John | 3 |  |  |  | 2 |  |
| Munday and Chettle | 1 |  |  |  | 1 |  |
| Porter, Henry | 1 |  |  |  |  |  |
| Shakespeare, William | 5 | 1 | 3 |  | 1 |  |
| Uncertain | 3 |  |  |  |  | 4 |
| Wilson et al. | 1 |  |  |  |  |  |

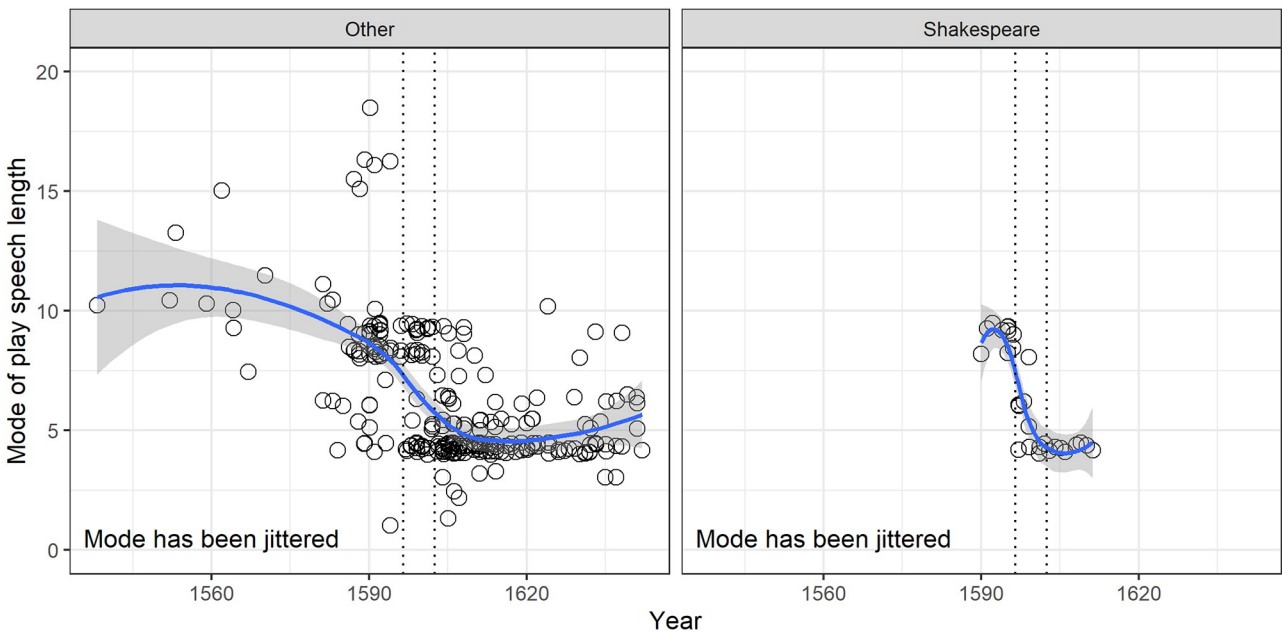

**Fig 11. (S1 Fig 13 in S1 File) The modes for speech lengths for Shakespeare's plays and for plays by others, with a lowess smoothed line, and the 95% confidence band.** The range for Shakespeare plays is 1590 to 1613; the range for plays by other authors is 1538 to 1642. The 1597 and 1602 boundaries are shown as dotted vertical lines.

is the only clear-cut pioneer in this group, though arguments might be made for Shakespeare also, with nine plays with shorter speech length modes and just one—*Henry V* (1599)—with a speech-length mode in the higher range. Most commonly, however, the more prolific authors had a mix of both lower- and higher-length speech modes.

The interest in speech length in early modern English plays began with Shakespeare [10]. In Fig 11 we plot the mode of speech length for Shakespeare plays with the mode of length for plays by his contemporaries.

The transition over time for Shakespeare's plays is very similar to that for other authors, with a sharp decline from predominant modes of 8 or 9 to a predominant mode of 4.

### 3.5 Analysis using only adult professional play types

The Adult Professional play type is well populated across the three time periods and the five genre groups. There is an opportunity therefore to run an analysis with only Adult Professional plays and exclude the influence of variations between Adult Professional, Boys' Professional and Other play types.

Firstly a model with genre, time period and their interaction was fitted, the interaction being not significant ($F_{(6,177.0)} = 0.30$, $p = .93$) so the model was simplified to a-main-effects-only model in which both genre and time period were significant ($F_{(4,182.9)} = 3.8$, $p = .005$ and $F_{(2,145.3)} = 40.0$, $p < .001$ respectively). Plots of the effects for the mode will be compared with that of the mean in the section 3.7.2 below.

### 3.6 Mean

**3.6.1 Mean speech lengths and genre, play type and time period.** As the distribution of speech lengths is right skewed, the mean as a measure of central tendency will be more sensitive to plays with longer speech lengths than the mode or the median.

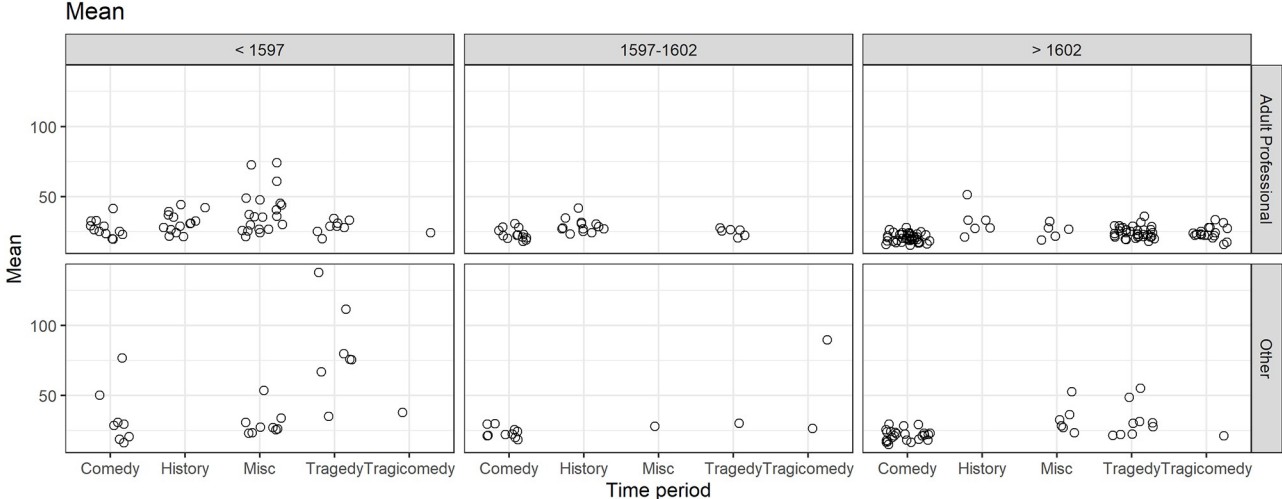

**Fig 12. (S1 Fig 14 in S1 File) Mean speech length of plays in genre, play type and period groupings.**

In Fig 12 the play means for combinations of genre, play type and time period are shown. The absence of history plays for the Other play type category is evident along with lower numbers of plays in the Other category (79/273 = 29%). Also the large spread in means for the Other play type for tragedies for the period prior to 1597 is noted. There were 9 plays with means greater than 70 (Table 4), 8 out of the 9 being prior to 1597.

A full factorial LMM was fitted to examine all combinations of genre, play type and time period. All interaction terms were significant suggesting a complex interpretation. The model output is available in S1 File and the EMMs and Confidence Intervals are plotted in S1 Fig 15 in S1 File. Apart from the missing combinations the most notable feature was in the Other play type category where there were two instances of substantially higher means, for Tragedy prior to 1597 and Tragicomedy in the 1597–1602 period. In the same figure the equivalent model for the mode is plotted and all categories were effectively equivalent, none of the interaction terms for the mode were significant. This is notable, showing how the mean as a summary measure is picking up some different features of plays than the mode.

**3.6.2 The mean of speech lengths and genre and time period for adult professional plays only.** Due to a number of missing combinations in the data complicating the interpretation, a simpler comparison was examined by excluding the Other play type category so

**Table 4. Plays with mean speech lengths greater than 70.**

| Author | Title | Date | Play Type | Genre | Mean of Speech Lengths |
|---|---|---|---|---|---|
| Norton and Sackville | Ferrex and Porrex | 1562 | Other | Tragedy | 137.7 |
| Daniel, Samuel | Cleopatra | 1593 | Other | Tragedy | 111.6 |
| Brandon, Samuel | Virtuous Octavia | 1598 | Other | Tragicomedy | 89.7 |
| Wilmot, Robert, et al. | Tancred and Gismund | 1567 | Other | Tragedy | 80.2 |
| Nashe, Thomas | Summer's Last Will and Testament | 1592 | Boys' Professional | Comedy | 77.1 |
| Kyd, Thomas | Cornelia | 1594 | Other | Tragedy | 75.8 |
| Sidney, Mary | Antonius | 1592 | Other | Tragedy | 75.5 |
| Uncertain | 1 Selimus | 1592 | Adult Professional | Heroical Romance | 74.4 |
| Peele, George | Battle of Alcazar | 1589 | Adult Professional | Foreign History | 72.7 |

that only Adult Professional plays were analysed. For the main-effects-only model genre and time period were significant ($F(4,187) = 11.3$, $p < .001$, $F(2, 187) = 9.8$, $p < .001$ respectively). Residual diagnostics were satisfactory for the mean model (S1 Fig 17 in S1 File) and the random effect diagnostics were not performed as the random effect SD estimate was zero.

In Fig 13 the patterns of the genre and time period effects were compared between the mean and the mode (from the model in section 3.5) for the Adult Professional plays. Note the large differences in the summary statistics for the distribution of speech lengths. The modes range from about 5 to 8 whereas the means range from about 24 to 34, having higher values and a larger range. As these are not directly comparable some kind of scaling would be needed to bring the measures to a common basis. Borrowing from the concept of standardized effect sizes relative to the uncertainty in the means, as shown by the Confidence Intervals, the separation between the genres of comedy, history, and miscellaneous is larger than that for the mode. The letter notations for assessing which pairs are significantly different in the mean plot help to understand this, the genre of comedy being significantly lower than that of both history and

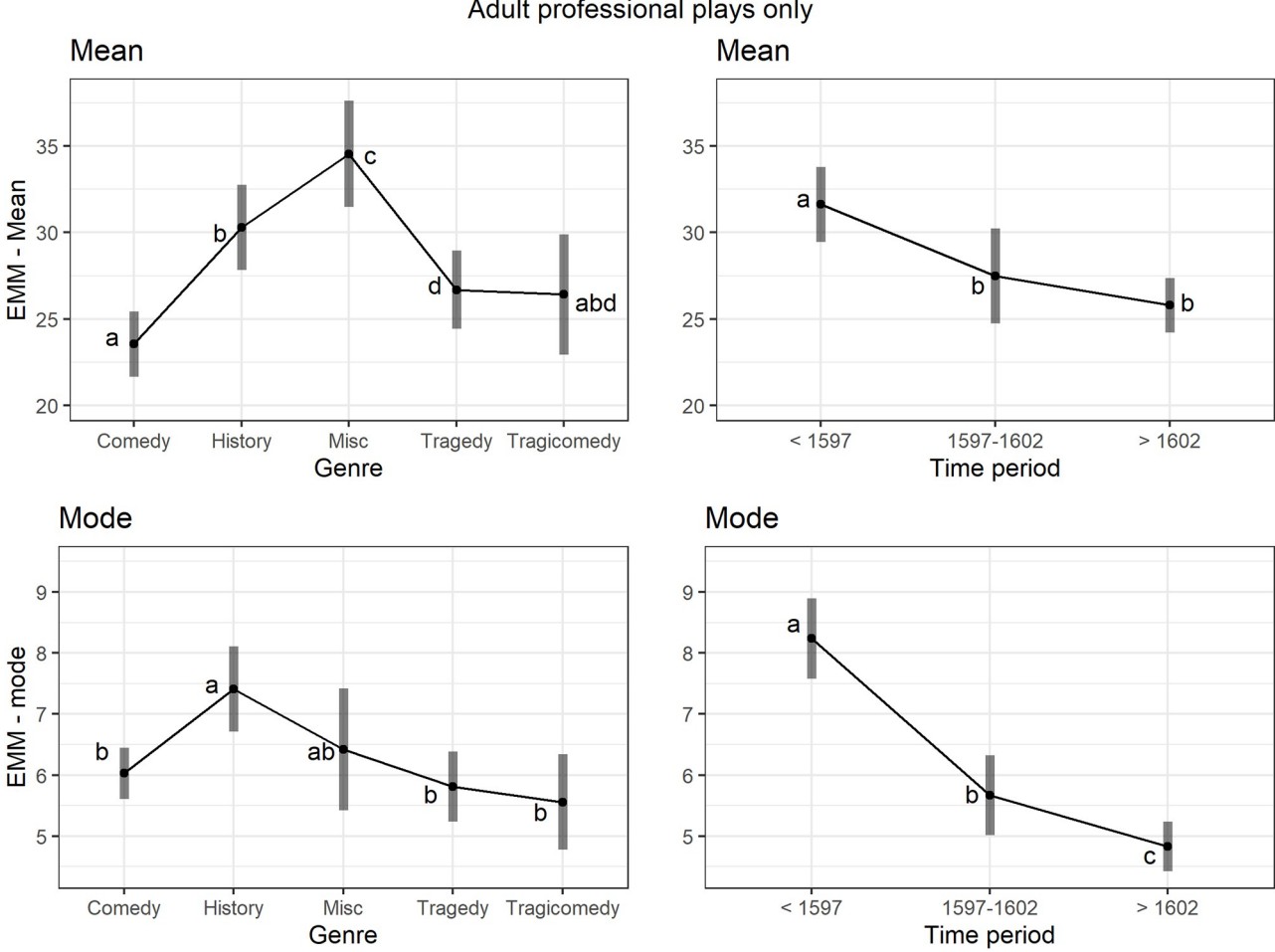

**Fig 13. (S1 Fig 16 in S1 File) Comparison of the mean and mode for the genre and time period effects for adult professional play types only.** EMMs and 95% Confidence Intervals from the main-effect-only models. Within each plot letters have been added to show which levels are statistically different. Categories with different letters are different; when letters are shared between groups the differences were not large enough to achieve significance.

miscellaneous for the mean and miscellaneous being significantly higher than that of history. In addition, for the mean the genre of comedy had the shortest speech length whereas for the mode the genres of tragedy and tragicomedy had lower modes. These differences in rank ordering are attributed to the greater sensitivity of mean to longer speech lengths compared to the mode. The mean and mode provide complementary views of the play speech length distributions and so each contributes to increased understanding of differences between the genres. For example with the EMM mean for comedy being lower than all the others it suggests that all speeches, short and long, were generally kept shorter in comedies compared to other genres for plays produced by adult professional companies. Also the time-period effect is shown more starkly with the mode, with the difference between the transition period 1597–1602 and the period after 1602 being more obvious than with the mean where the last two periods are not significantly different.

### 3.7 Median

Although the median is typically intermediate in position between the mode and the mean for skewed distributions, fitting the equivalent main-effects-model to that for the mean gave very similar results. The statistical significance and EMM patterns were very similar to that of the mean. The statistical details are available in S1 File and the patterns compared in S1 Fig 19 in S1 File.

## 4 Discussion

### 4.1 Importance of effects

The LMM analysis provided a way to rank the importance of each of the variables modelled. The largest impact was attributed to change over the time period studied. Using the LMM model based EMMs the mean of the mode decreased from 8.5 prior to 1597 to 4.8 after 1602, a change of 3.7. The genre effect was the next most important with the largest difference in EMMs being between the highest for historical plays with a mean mode of 7.5 to the lowest for the tragedy and tragicomedies, about 5.8, which is a difference 1.7 and less than half of the change over time. Differences between authors were of a similar size to that of genre, however as author was modelled as a random effect rather than fixed the method of assessment was different. Using the normal distribution as a model for the variability between authors 95% of authors would be being expected to vary in the mean mode over a range of 1.4. The two other variables examined as part of the modelling process play type (Adult Professional compared to Boys' Professional) and proportion of verse in a play were not statistically significant, suggesting small effects, perhaps close to zero.

One other effect is noteworthy: the proportion of verse in a play. Although this variable was not significant in the LMM mode analysis it had a noticeable effect on the shape of the distributions of the pre-1597 data. At high verse proportion percentages the distribution of speech lengths had multiple peaks (modes) suggesting that the verse structure of the writing was important. However after 1602 this effect had largely disappeared.

Examining the mean instead of the mode of play speech length using only adult professional plays showed similar patterns to that of the mode. However, compared to the uncertainty in the EMM estimates the time-period effect was relatively smaller and the genre effect was relatively larger with the largest difference being between historical plays and comedies. These differences probably reflected the greater sensitivity of the mean to longer speech lengths compared to the mode.

## 4.2 Speech lengths as experienced by audiences

Patterns of variation in speech length are of interest to literary scholars because of their role in the overall effect of plays on audiences and readers. To understand this better we can examine some examples of various-length speeches in context.

We can compare a passage from Shakespeare's play *The comedy of errors* (1594) and another from the same playwright's *All's well that ends well* (1603) to see what happens in two passages by the same playwright, in the same genre, but with one some years further on in the evolution of English Renaissance drama, and on the far side of the change to shorter speeches. (In the quotations that follow, speeches counted as four words are in bold type).

*The comedy of errors* (1594) [34] III.ii.54-65

LUCIANA What, are you mad that you do reason so?
ANTIPHOLUS OF SYRACUSE Not mad, but mated—how, I do not know.
LUCIANA It is a fault that springeth from your eye.
ANTIPHOLUS OF SYRACUSE For gazing on your beams, fair sun, being by.
LUCIANA Gaze when you should, and that will clear your sight.
ANTIPHOLUS OF SYRACUSE As good to wink, sweet love, as look on night.
LUCIANA Why call you me "love"? Call my sister so.
ANTIPHOLUS OF SYRACUSE Thy sister's sister.
LUCIANA **That's my sister**.
ANTIPHOLUS OF SYRACUSE No,
It is thyself, mine own self's better part,
Mine eye's clear eye, my dear heart's dearer heart,
My food, my fortune, and my sweet hope's aim,
My sole Earth's heaven, and my heaven's claim.
LUCIANA All this my sister is, or else should be.

In the texts used for the data sets for this study, as discussed in the Materials and Methods section above, elisions are expanded for counting purposes, so that "That's my sister" is changed to "That is my sister." In our counts these eleven speeches are (9,9,9,9,10,10,10,3,4,35,9). They are all in verse, in iambic pentameter, with one line split three ways (Thy sister's sister. / That's my sister. / No. . .). The mode for speech length in this play is 9 (76 of 609 speeches, or 12.5%, are of this length; there are 95 unique speech lengths, ranging from 1 to 493 words). In the passage quoted here, there is a mixture of short and long speeches, with one-line speeches of 9 or 10 words accounting for five of them. The full one-line speeches give room for some complexity and detail—"What, are you mad that you do reason so?"—but do not linger long before there is a retort or new development from the other speaker. We cannot claim that this passage is typical of the speech-rhythm of the play, but it does give a sense of what a modal speech length of nine would mean: nothing unduly hurried; but not too static either.

We can contrast this with *All's well that ends well* (1603). In this case there is a double mode, with 61 speeches each for speech lengths 4 and 5. There are 944 speeches in all, so these speech lengths each account for 6.5% of the speeches, and together for 13%. There are 115 different speech lengths in all, from 1 to 252 words. Consider the following passage from Act 1 Scene 1, in prose this time.

*All's well that ends well* (1603) [35] I.i.162-185

PAROLLES Little Helen, farewell; if I can remember thee, I will think of thee at court.
HELENA Monsieur Parolles, you were born under a charitable star.
PAROLLES Under Mars, I.
HELENA I especially think, *under* Mars.

PAROLLES Why "*under* Mars"?

HELENA The wars have so kept you under that you must needs be born under Mars.

PAROLLES **When he was predominant**.

HELENA When he was retrograde, I think, rather.

PAROLLES **Why think you so?**

HELENA You go so much backward when you fight.

PAROLLES **That's for advantage**.

HELENA So is running away, when fear proposes the safety; but the composition that your valour and fear makes in you is a virtue of a good wing, and I like the wear well.

PAROLLES I am so full of businesses, I cannot answer thee acutely. I will return perfect courtier; in the which, my instruction shall serve to naturalize thee, so thou wilt be capable of a courtier's counsel and understand what advice shall thrust upon thee; else thou diest in thine unthankfulness, and thine ignorance makes thee away. Farewell. When thou hast leisure, say thy prayers; when thou hast none, remember thy friends; get thee a good husband, and use him as he uses thee. So farewell.

The speech lengths for these thirteen speeches are (15,9,3,5,3,15,4,7,4,8,4,33,84). The alternation is on the whole more rapid than in the previous passage, though the short exchanges ("Why think you so?" / "You go so much backward when you fight." / "That's for advantage.") are mixed with longer ones. There is more variation in speech length. In the play as a whole, the mode for speech length is 4 words.

## 4.3 Stylistics of speech lengths

The change in most common speech length from 9 words to 4 words is a move from the predominance of full statements to the predominance of pithy summaries. In stylistic terms this is a move from informational and self-sufficient communication, in which a context is provided, to involved exchanges which assume shared knowledge. The first has high informational density and precise content, the second has more interactional, affective and generalised content. The first is more characteristic of writing and the second of speech.

Longer speeches make it possible and likely that the speaker will add more detail and specificity, so that statements require less completion by the audience. Short speeches tend to include more references to persons and things known to speaker and audience through pronouns and deictics. This contrast corresponds to the first factor in Biber's factor analysis of a range of documents [36]—informational versus involved styles. Writing tends to have more of the first, since there is the opportunity to elaborate and edit, and the assumed audience is generally not present, and speech has more of the second, since the statement has to be completed in real time and the audience is generally present.

The underlying question is why playwrights changed in their practice. Here we can only speculate. One plausible hypothesis is that they learned progressively from one another how to represent more closely the speech lengths of everyday exchanges, and found that audiences responded well to these. As the heritage of early modern English drama grew, playwrights came to understand the special properties of the medium better and their styles moved away from an allegiance to writing and towards the dynamics of everyday speech. George T Wright, already quoted, discussing the increase in shared lines in Shakespeare's dramatic verse over his career, offers an attractive explanation for Shakespeare's change of practice: lines shared between characters "suggest the casualness of natural conversation" and Shakespeare included more as time went on from a "wish to present credible and credibly various language on the stage" [12 p140, p120]. The same might be said about the change to shorter speeches in general in the English drama of this period.

A change in the direction of closeness to natural speech has also been observed in Shake-speare's metrical practice. Wright showed that in his early plays Shakespeare tended to finish each sentence at the end of a verse line and hence begin the next sentence at the start of a verse line, and that in multi-line sentences the ends of lines tended to align with the ends of phrases and clauses [12]. In this regard his early plays were typical of the collective practice. But Shake-speare led the field in certain connected deviations from this practice. Ants Oras traced the way that all dramatists' plays drifted in their most commonly chosen position for a midline break - - such as a caesura, strong punctuation, or the splitting of a verse line between two speakers - - from the first half of the line (especially after the fourth syllable) when Shake-speare's career began in the late 1580s to the second half of the line (especially after the sixth syllable) by the time Shakespeare's career ended around 1613 [37]. Starting the second half of a line later tended to make phrases and clauses overrun the end of the line, and this enjambment diminished the noticeability of line endings in performance, and hence their significance to writers and performers.

The collective change to shorter speeches can also be seen as driven by fashion, as a structural variation which practitioners adopted as an attractive novelty. There are other examples of change spreading quickly across the English theatre of the time. The division of plays into five acts was introduced in the second decade of the seventeenth century and quickly became uni-versal. Plays performed indoors were illuminated by candles, and these would need periodic attention to put out the ones that were smoking, relight those that had gone out, and where nec-essary to trim the wicks. Indoor venues such as the Blackfriars theatre always punctuated perfor-mances with four intervals during which the candles were tended, and at the Blackfriars the audience were kept entertained by a group of musicians visible in the balcony over the stage. Outdoor venues such as the Globe theatre used natural daylight and, having no need for inter-vals, performed plays as a continuous sequence of scenes. In 1610, Shakespeare's company, the King's Men, began to play at the indoor Blackfriars theatre in the winter while continuing to use their outdoor Globe theatre in the summer. The company decided to regularize the practices at the two venues, moving the Globe's musicians' room from inside the tiring house to the stage balcony, as at the Blackfriars, punctuating Globe performances with four musical intervals even though there were no candles to tend [38]. The King's Men presumably found it convenient to perform the same way outdoors as indoors. Punctuating outdoor performances with four unnecessary intervals was quickly emulated by other companies that had no indoor theatres [38]. Every one of the 245 extant plays (whether printed or manuscript) from 1616 to 1642 writ-ten for London companies by 56 different authors is divided into five acts, showing that by 1616 the transition to universal use of intervals was complete [39 p4]. Thus we see that an innovation can rapidly spread across an entire industry, even when there is no practical reason for it.

Another way to think of a change in speech length is offered by the Russian literary scholar Boris Yarkho (1889–1942) [40]. Yarkho may well have been the first to consider the length of speeches in plays in a systematic way. He proposed an "index of liveliness," being the ratio of the number of speeches to the number of lines in a play and calculated this index for the come-dies and tragedies of the seventeenth-century French playwright Pierre Corneille, comedies having a higher index, and thus a shorter mean speech length. The move from a mode of nine words to a mode of four corresponds to a shortening of the average speech, and thus a move to more lively drama in Yarkho's terms.

## 4.4 Implications for literary studies

We hope that this demonstration of the application of Linear Mixed Models to stylistic ques-tions will prompt other studies in stylistics using this method. LMM can account for the

influence of multiple variables on an outcome variable like speech length in a systematic way, allowing the researcher to determine whether a given factor like time period is indeed dominant. The concept of random effects is useful in the typical situation of literary studies where samples are not elicited by a process ensuring balanced stratification but derive from a dataset with inbuilt limitations and biases, such as the surviving written texts from a historical tradition.

In terms of implications for the specific field of studies of Shakespeare and his contemporaries three aspects stand out from the speech-length patterns we have discussed.

The patterns provide a new focus on shorter speeches, which turn out to dominate the speech-length distribution. This offers an important context for work on dramatic dialogue analysing briefer exchanges [41, 42], which has emerged alongside the older tradition of commentary on longer speeches, such as soliloquies and speeches reporting off-stage events, declaring sentiments, or attempting to persuade. The findings also provide an important and well-founded context of change over time for these studies of dialogic interaction.

The second is that the direction of the change, towards dialogue with shorter speeches, seems intuitively to be an adaptation to a closer modelling of natural speech. In their book on stylistic patterns in early modern English drama, Craig and Greatley-Hirsch [1] described a collective and progressive drift in early modern English drama over the years 1585 to 1624 towards more informal and interactional dialogue, in that case based on frequencies of function words. This is consistent with the change in speech-length modes. Craig and Greatley-Hirsch did not argue that this is an improvement. One might well prefer early drama, with longer speeches, to later examples, but nevertheless the writers seem to have moved collectively towards verisimilitude, so that once the late style is experienced, the earlier one seems more removed from everyday reality.

The third is the reminder that all these dramatists worked under invisible constraints. Any playwright who wrote before 1597 tended to make the nine-word speech the most common type; any playwright who wrote after 1602 tended to make the four-line speech the most common type. This is in a sense no surprise. Fashions change, and writers like everyone else take advantage of the new possibilities created by the works of their immediate predecessors, and respond to what audiences have learned to want. Yet this influence from a common drift in style is easy to forget when focusing on a single work or even a single writer, and broad-based quantitative studies help in re-balancing commentary towards this wider collective context, towards remembering that a play's place in the chronological sequence is a factor in its style, alongside as the more familiar considerations like its author. It is important to know that *All's well that ends well* was first performed in 1603, as well as the fact that it was written by William Shakespeare.

## Supporting information

**S1 File. File with R script, output and text explanations.**
(HTML)

**S2 File. File with R script, output and text explanations.**
(HTML)

**S3 File. File with R script, output and text explanations.**
(HTML)

**S4 File. Data file of all speech length values from all 275 plays.**
(TXT)

**S5 File. Summary data file containing play mode, median and mean of speech length in words, with play metadata.**
(TXT)

**S6 File. R Markdown file to reproduce the output in S1 File.**
(RMD)

**S7 File. R Markdown file to produce the output in S2 File.**
(RMD)

**S8 File. R Markdown file to produce the output in S3 File.**
(RMD)

## Author Contributions

**Conceptualization:** Kim Colyvas, Gabriel Egan, Hugh Craig.

**Data curation:** Kim Colyvas, Gabriel Egan, Hugh Craig.

**Formal analysis:** Kim Colyvas, Gabriel Egan.

**Funding acquisition:** Gabriel Egan, Hugh Craig.

**Investigation:** Kim Colyvas, Gabriel Egan, Hugh Craig.

**Methodology:** Kim Colyvas, Gabriel Egan, Hugh Craig.

**Project administration:** Kim Colyvas, Hugh Craig.

**Resources:** Kim Colyvas, Gabriel Egan, Hugh Craig.

**Software:** Kim Colyvas.

**Validation:** Kim Colyvas.

**Writing – original draft:** Kim Colyvas, Gabriel Egan, Hugh Craig.

**Writing – review & editing:** Kim Colyvas, Gabriel Egan, Hugh Craig.

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
