## [Editor Report · Decision Letter 0]

8 Jul 2022

PONE-D-22-17959Changes in the length of speeches in English plays (1538-1642): a linear mixed models approachPLOS ONE

Dear Dr. Hugh Craig,

Thank you for submitting your manuscript to PLOS ONE. After careful consideration, we feel that it has merit but does not fully meet PLOS ONE’s publication criteria as it currently stands. Therefore, we invite you to submit a revised version of the manuscript that addresses the points raised during the review process.

I found the work interesting and worth of publication. However, before sending this paper out to reviewers I read it myself and came to conclusion that some critical modifications and revisions are to be made before giving the paper consideration for publication.

Major concerns:

The innovative centerpiece of the paper is the application of the linear mixed model (LME) to demonstrate that the reduction of the average number of words per sentence in the play is robust with respect to other contributing factors, such as authorship, genre, theater, etc. To my surprise, I did not find any report of the LME model these claims are based on. This model along with all parameters and its statistical characteristics must be displayed in the main body of the paper accompanied with detailed explanation. The authors must explain all variables and justify the choice of the random effect. Note that the discussion of the model in SI part is not sufficient. Moreover, it seems that the authors present many models in SI, but it’s unclear what model is the final for interpreting the results and drawing conclusions in the abstract. Generally, I would like to have more detailed description of the methodology – the driving force of the results of the work.The nonlinear curves derived through nonparametric estimation via splines displayed in Fig. 1, 5, etc. are not justified. They look to me as the result of overfitting – a typical artifact unless they can be explained and tested through extensive simulations. Instead, just using a simpler but more robust two-segment regression would be more trustful.The authors should explain or at least offer some hypothesis why sentences became dramatically shorter within a 5-year period. What happened? That would considerably increase the visibility of the work.It is established in statistics to use Poisson distribution for counts such as the number of words in the sentence. I suggest reconsidering the model using log link in the GLM mixed model. Alternatively, the authors may work with log-transformed counts. This approach will substantially reduce the effect of outliers.I see lack of statistical significance testing and reporting p-values when comparing the results from two time periods. These p-values should be displayed right on the figures.  

Minor concerns:

Replace “English plays (1538-1642)” with “…plays of William Shakespeare and his contemporaries…” I believe that the new title is catchier and will attract larger readership.“In this paper we aim to model speech length variation in these plays as comprehensively as possible” is not specific enough. Explain exactly how your work is different from others. Using mixed model?You use the mode as the central tendency measure but mixed model assumes that the central tendency measure is the mean. I see a contradiction.136: “…with mixed modes..” Is it a typo? Why modes? Shouldn’t be models?

We look forward to receiving your revised manuscript.

Kind regards,

Eugene Demidenko, Ph.D.

Academic Editor

PLOS ONE

2. Please ensure that you refer to Figure 7 in your text as, if accepted, production will need this reference to link the reader to the figure.

3. We note you have included a table to which you do not refer in the text of your manuscript. Please ensure that you refer to Table 4 in your text; if accepted, production will need this reference to link the reader to the Table.

5. Please upload a copy of all your Supporting Information Figure which you refer to in your text.

---

## [Author Response · Author response to Decision Letter 0]

5 Dec 2022

Major concerns:

1. The innovative centrepiece of the paper is the application of the linear mixed model (LME) to demonstrate that the reduction of the average number of words per sentence in the play is robust with respect to other contributing factors, such as authorship, genre, theater, etc. To my surprise, I did not find any report of the LME model these claims are based on. This model along with all parameters and its statistical characteristics must be displayed in the main body of the paper accompanied with detailed explanation. The authors must explain all variables and justify the choice of the random effect. Note that the discussion of the model in SI part is not sufficient. Moreover, it seems that the authors present many models in SI, but it’s unclear what model is the final for interpreting the results and drawing conclusions in the abstract. Generally, I would like to have more detailed description of the methodology – the driving force of the results of the work.

Response

Requested to provide more detail about the methodology and the LME model.

- Mixed model parameter and statistical characteristics with detailed explanation

A section has been added following the statistical methods section describing aspects of the mixed model. Keeping in mind the likely readership of the paper the mathematical components were kept to a minimum, trying to focus more on the concepts.

- Explain all variables

Section 2 has been reorganised and amplified to describe each variable in turn.

- Justify author as a random effect

Justification was added to the new mixed model section and supported by showing in section 3.4 Speech length by author how the author random effect could be applied to make inference about the wider population of authors. 

- Many models have been fitted and 

what model is the final for interpreting the results and drawing conclusions in the abstract

The problem of many models in S1 has been reduced considerably. Most of the alternative models were moved to S2 as part of the investigation comparing the normal model and the Poisson. Resulting from that one final model in S1 was used as the foundation for reporting results in the manuscript. Small variations on that model in S1 were minor (test interactions, repeat with data reduced to one play type) and based on the final model. 

2. The nonlinear curves derived through nonparametric estimation via splines displayed in Fig. 1, 5, etc. are not justified. They look to me as the result of overfitting – a typical artifact unless they can be explained and tested through extensive simulations. Instead, just using a simpler but more robust two-segment regression would be more trustful.

Response

These kind of smoothing curves are to assist the viewer in detecting patterns in the data. The one chosen (lowess) is widely used for this purpose and has reasonable properties not being too reactive to the data. A less responsive smoothing method has been chosen to replace it based on Generalised Additive Models (GAM). Again, the only purpose was to guide the reader as to any general pattens and pick up any curvature that might be present in the data as opposed to forcing a linear relationship interpretation. Any statements about what might be significant would be supported by formal statistical tests after clues from the plots alerted the investigator to possible patterns in the data.

3. The authors should explain or at least offer some hypothesis why sentences became dramatically shorter within a 5-year period. What happened? That would considerably increase the visibility of the work.

Response

We have amplified the section on the literary and theatrical context for the observed changes (4.2) to provide hypotheses for causation, i.e. the authors’ wish to make onstage dialogue more like natural speech; a collective adoption of a new fashion; and an authorial wish for more lively dramatic dialogue.

4. It is established in statistics to use Poisson distribution for counts such as the number of words in the sentence. I suggest reconsidering the model using log link in the GLM mixed model. Alternatively, the authors may work with log-transformed counts. This approach will substantially reduce the effect of outliers.

Response

The Poisson distribution would have been most directly applicable to the original speech length data where the measure was a count. However, the data forming most of the analyses was based on summary statistics from each play, mode, mean and median. Summary statistics would be expected to have different properties to the original measure due to condensing the information in the whole distribution into a single value. 

The suggestion to use the Poisson distribution was not unreasonable however due to the discrete nature of the data so Poisson models were evaluated against the normal model. Initially it appeared the Poisson was the better choice. In seeking to understand why it was better some important issues were discovered to do with influential observations and non-constant variance in the residuals. The investigation developed into a very substantial side investigation that is documented in an additional supplementary file (S2). Eventually the normal distribution was found to be a better choice than the Poisson, but the quality of the normal model was improved over the original version because of what was found. A summary describing more details about this and what was done provided is at the beginning of S2.

5. I see lack of statistical significance testing and reporting p-values when comparing the results from two time periods. These p-values should be displayed right on the figures.

Response

Letter codes have been added to the plots with CIs to indicate which categories are statistically significantly different. The use of these in interpretation is supported by text in the figure captions and the manuscript text.

Minor concerns:

1. Replace “English plays (1538-1642)” with “…plays of William Shakespeare and his contemporaries…” I believe that the new title is catchier and will attract larger readership.

Response

Good suggestion, we have made the change.

2. “In this paper we aim to model speech length variation in these plays as comprehensively as possible” is not specific enough. Explain exactly how your work is different from others. Using mixed model?

Response

We have added several sentences at the end of the Introduction to detail the innovations in the article.

3. You use the mode as the central tendency measure but mixed model assumes that the central tendency measure is the mean. I see a contradiction.

Response

There is no contradiction, as explained above the mode is a summary measure of a distribution, in other words it is just a derived variable based on measurements. When we use another summary statistic, the mean, to summarise a general tendency or pattern for the mode from different plays, it is no different to using the mean to summarise a range of values based on any variable. The question we are answering is something like this. The process of making plays produces a range of speech lengths and when summarised by modes is not the same for each play. So how do we summarise the distribution of modes across many different plays. The mean is a sensible and natural option, but others could have been chosen, even the mode! But statistical theory and methods for the mean are much more developed, available and understood so the mean was the preferred choice.

4. 136: “…with mixed modes..” Is it a typo? Why modes? Shouldn’t be models?

Response

Modes was correct, the idea we were trying to explain was that between the early and final period there was a transition period in which both kinds of plays were being produced, those with high modes and those with low. 

The word mixed was been expanded now reading

“1597-1602 plays (with a mixture of high and low modes)”

Approach

---

## [Decision Letter · Decision Letter 1]

11 Jan 2023

PONE-D-22-17959R1Changes in the length of speeches in the plays of William Shakespeare and his contemporaries: a mixed models approachPLOS ONE

Dear Dr. Craig,

Thank you for submitting your manuscript to PLOS ONE. After careful consideration, we feel that it has merit but does not fully meet PLOS ONE’s publication criteria as it currently stands. Therefore, we invite you to submit a revised version of the manuscript that addresses the points raised during the review process.

Two qualified reviewers provided comments that must be fully addressed in the point-by-point fashion. 

We look forward to receiving your revised manuscript.

Kind regards,

Eugene Demidenko, Ph.D.

Academic Editor

PLOS ONE

Journal Requirements:

Reviewers' comments:

Reviewer's Responses to Questions

**Comments to the Author**

1. If the authors have adequately addressed your comments raised in a previous round of review and you feel that this manuscript is now acceptable for publication, you may indicate that here to bypass the “Comments to the Author” section, enter your conflict of interest statement in the “Confidential to Editor” section, and submit your "Accept" recommendation.

Reviewer #1: All comments have been addressed

Reviewer #2: (No Response)

2. Is the manuscript technically sound, and do the data support the conclusions?

Reviewer #1: Partly

Reviewer #2: Yes

3. Has the statistical analysis been performed appropriately and rigorously? 

Reviewer #1: No

Reviewer #2: Yes

4. Have the authors made all data underlying the findings in their manuscript fully available?

Reviewer #1: Yes

Reviewer #2: Yes

5. Is the manuscript presented in an intelligible fashion and written in standard English?

Reviewer #1: Yes

Reviewer #2: Yes

6. Review Comments to the Author

Reviewer #1: Comments

This manuscript demonstrates an interesting attempt of applying statistical methods to analyzing the development trends in literary studies. Informative research findings include a shift towards shorter speeches used in plays and the possible influence of writing fashion. My suggestions focus on the statistical analysis section of this study:

1. “The length of words in speeches”, as was mentioned on Page 1, is the outcome variable/independent variable of the study. The authors have endeavored to provide a clear definition of “speech” on Page 3. Personally I think the readers would benefit from an example from the play scripts. “Speech” is often measured as the number of syllables between identifiable pauses in applied linguistics, while the number of words is the major index for evaluating speech length. In the literature review section, 2007 MacDonald P Jackson adopted a different system in calculating speech length—3-6 words/3-10 words were categorized as a speech. The authors have also made the argument of why syllabification is not used in this study on Page 8. It would help the readers, however, to dissolve possible doubts and keep the measurement of speech consistent if explanations appear at an earlier place in this manuscript.

2. I appreciate the detailed explanation of statistical methods from Page 11 to Page 14. However, this section could be reduced to some extent, as readers who have background knowledge in quantitative analysis usually understand how linear mixed models work. The selection of variables, or the identification of key predictors for speech length, requires a large amount of justification. This is in connection with the literature review section. I would recommend that the researchers dedicate more efforts to investigating previous studies discussing possible predictors of speech length, such as authorship, genre, etc. It is highly possible that not a lot of quantitative studies could be found, as statistical analysis might not have been the main research paradigm in literary studies. However, researchers’ discussion of possible factors would still help answer the major question of this study: which factors need to be included (or excluded) from the linear mixed model and why?

3. As for the discussion and implication section, my suggestions originate more from an applied linguistic approach. In addition to identifying significant predictors, the function of linear mixed models also lies in sorting out the most powerful factors rendering the final outcome variable. In the discussion section, instead of starting with an audience perspective, the authors could briefly report the coefficients in the model and discuss significant predictors and most influential predictors, which might be corroborated by the scripts used as examples.

It is exciting to see quantitative methods are also playing a role in literary research, and interdisciplinary studies were conducted in-depth with the help of statistics. The innovative methodology deserves a more prominent place in the implication part, which hopefully would be assist researchers with similar needs in their academic inquires.

Reviewer #2: Manuscript Number: PONE-D-22-17959R1

Manuscript Title: Changes in the length of speeches in the plays of William Shakespeare and his contemporaries: a mixed models approach

The article does a good job of establishing its findings in a succinct and easy-to-understand manner. However, there were a few issues that need to be revised. There are as follows:

Major issues:

While the overall quality of the article is good and it is scientifically sound, quite a few claims made in the article are unsubstantiated.

For instance:- the sentence ‘The change in speech lengths is part of a wider collective drift in the plays towards liveliness and verisimilitude and is evidence of a hitherto hidden constraint on the playwrights.’ might be accurate, it would do well to establish this with a few instances of this so-called ‘hidden constraint’ or ‘hidden patterns’ as mentioned in section 1.

Similarly, the phrase ‘publishers who applied other equally well-established typographical conventions to mark off the speeches when printing plays from manuscripts.’ In section 1 would also do well by adding a few examples of these ‘typographical conventions’ to substantiate the claim of the sentence. (i.e. everyone knows there are certain ‘typographical conventions to mark off speeches’, but in a scientific article, these do need to be explicitly mentioned, even if in passing.)

The sentence ‘Comparatively long speeches, such as soliloquies and speeches reporting off-stage events, declaring sentiments, or attempting to persuade, have attracted the most commentary to date, so we might assume this sort of length is dominant.’ In section 1 is another example of a sentence making a sweeping remark without substantiating it. (While it be accurate that longer speeches attract more commentary, in a scientific article you need to back that up with facts and proof, either referencing from where you derive this information, or making this analysis yourself from the data you have.)

There are quite a few sentences in the article that share this issue, and they stick out like a sore thumb in an otherwise well-writ and researched article. I would suggest that the authors take some time to go through the article to either substantiate (with valid references), or get rid of, unsubstantiated claims.

Minor issues:

The article is afflicted with several grammatical and spelling issues, which is unbecoming of a scientific article, specially one that deals with William Shakespeare’s plays. It would be better to go through the article closely and remove any syntactic inaccuracies. A few examples of these are :

in section 1.2, ‘ Ilsemann observed that in these plays the commonest speech length …’

shouldn’t it be ‘most common’ instead of ‘commonest’, as it is much widely used and also avoids ambiguity? In a similar fashion, in the same section, the sentence ‘Lines shared between speakers were rare in Shakespeare’s earliest plays, making up two per cent or so of the total, but became more common in his middle period, and commoner still at the end, with the last plays.’ Makes use of the word ‘commoner’, rather than ‘even more common’. The word ‘commoner’, specially in an article pertaining to Shakespearean plays, could easily be misinterpreted as ‘a common person, as distinguished from one with rank, status, etc.’ and introduce ambiguity in a otherwise lucid and well-articulated article.

In the abstract, the sentence ‘We confirm that that the mode …’ falls prey to one of the more common errors that occur while typing or writing anything in English. The dreaded ‘that that’ in the abstract of an scientific article is disappointing.

I would suggest the authors to comb through the article and rid it of these minor issues before the article is ready for publication.

7. PLOS authors have the option to publish the peer review history of their article (what does this mean?). If published, this will include your full peer review and any attached files.

Reviewer #1: No

Reviewer #2: **Yes: **Abhinav Jha

---

## [Author Response · Author response to Decision Letter 1]

10 Feb 2023

Reviewer # 1: Comments

This manuscript demonstrates an interesting attempt of applying statistical methods to analyzing the development trends in literary studies. Informative research findings include a shift towards shorter speeches used in plays and the possible influence of writing fashion. My suggestions focus on the statistical analysis section of this study:

1. “The length of words in speeches”, as was mentioned on Page 1, is the outcome variable/independent variable of the study. The authors have endeavored to provide a clear definition of “speech” on Page 3. Personally I think the readers would benefit from an example from the play scripts. “Speech” is often measured as the number of syllables between identifiable pauses in applied linguistics, while the number of words is the major index for evaluating speech length. In the literature review section, 2007 MacDonald P Jackson adopted a different system in calculating speech length—3-6 words/3-10 words were categorized as a speech. The authors have also made the argument of why syllabification is not used in this study on Page 8. It would help the readers, however, to dissolve possible doubts and keep the measurement of speech consistent if explanations appear at an earlier place in this manuscript.

>We have added some detail on our definition of a speech and given an example, as suggested by 

>the reviewer, at ll. 55-65. We now make it clear that Jackson 2007 was selecting from among 

>available speeches, rather than adopting a different system to calculate speech length, as the 

>reviewer wrongly understood, at ll. 131-3. As the reviewer recommended, we have inserted a 

>brief preliminary defence of the choice to count words rather than syllables, at ll. 87-92.

2. I appreciate the detailed explanation of statistical methods from Page 11 to Page 14. However, this section could be reduced to some extent, as readers who have background knowledge in quantitative analysis usually understand how linear mixed models work. 

>We have interpreted this to mean mostly the mixed model theory related description rather than 

>the methods used for the model fitting and have trimmed this section. 43 lines are now reduced 

>to 24 and 628 words reduced to 342. See ll. 369-418.

The selection of variables, or the identification of key predictors for speech length, requires a large amount of justification. This is in connection with the literature review section. I would recommend that the researchers dedicate more efforts to investigating previous studies discussing possible predictors of speech length, such as authorship, genre, etc. It is highly possible that not a lot of quantitative studies could be found, as statistical analysis might not have been the main research paradigm in literary studies. However, researchers’ discussion of possible factors would still help answer the major question of this study: which factors need to be included (or excluded) from the linear mixed model and why?

>We have revised Section 2.2.2 (ll. 224-287), now describing more fully the motivation for the 

>choice of each variable.

3. As for the discussion and implication section, my suggestions originate more from an applied linguistic approach. In addition to identifying significant predictors, the function of linear mixed models also lies in sorting out the most powerful factors rendering the final outcome variable. In the discussion section, instead of starting with an audience perspective, the authors could briefly report the coefficients in the model and discuss significant predictors and most influential predictors, which might be corroborated by the scripts used as examples.

>We have added a new section, 4.1, “Importance of effects”, of about a page, to respond to this 

>suggestion, and renumbered the following sections to match.

It is exciting to see quantitative methods are also playing a role in literary research, and interdisciplinary studies were conducted in-depth with the help of statistics. The innovative methodology deserves a more prominent place in the implication part, which hopefully would be assist researchers with similar needs in their academic inquires.

>The methodology is now cited as one of the important implications of the study at the beginning 

>of Section 4.4, ll. 991-1001.

Reviewer # 2: Manuscript Number: PONE-D-22-17959R1

Manuscript Title: Changes in the length of speeches in the plays of William Shakespeare and his contemporaries: a mixed models approach

The article does a good job of establishing its findings in a succinct and easy-to-understand manner. However, there were a few issues that need to be revised. There are as follows:

Major issues:

While the overall quality of the article is good and it is scientifically sound, quite a few claims made in the article are unsubstantiated.

For instance:- the sentence ‘The change in speech lengths is part of a wider collective drift in the plays towards liveliness and verisimilitude and is evidence of a hitherto hidden constraint on the playwrights.’ might be accurate, it would do well to establish this with a few instances of this so-called ‘hidden constraint’ or ‘hidden patterns’ as mentioned in section 1.

>We have clarified with more explanation at ll. 35-6.

Similarly, the phrase ‘publishers who applied other equally well-established typographical conventions to mark off the speeches when printing plays from manuscripts.’ In section 1 would also do well by adding a few examples of these ‘typographical conventions’ to substantiate the claim of the sentence. (i.e. everyone knows there are certain ‘typographical conventions to mark off speeches’, but in a scientific article, these do need to be explicitly mentioned, even if in passing.)

>Examples of typographical and manuscript conventions to mark off speeches are now given at ll. 

>72-75.

The sentence ‘Comparatively long speeches, such as soliloquies and speeches reporting off-stage events, declaring sentiments, or attempting to persuade, have attracted the most commentary to date, so we might assume this sort of length is dominant.’ In section 1 is another example of a sentence making a sweeping remark without substantiating it. (While it be accurate that longer speeches attract more commentary, in a scientific article you need to back that up with facts and proof, either referencing from where you derive this information, or making this analysis yourself from the data you have.)

>We have modified the text at ll. 78-79 to make the remark less sweeping and when returning to 

>the same issue in Section 4.4 have re-framed the context and added two references to 

>substantiate the claims (ll. 1003-10).

There are quite a few sentences in the article that share this issue, and they stick out like a sore thumb in an otherwise well-writ and researched article. I would suggest that the authors take some time to go through the article to either substantiate (with valid references), or get rid of, unsubstantiated claims.

>We have added references at strategic points. There are now 42 references compared to 29 in 

>the earlier version.

Minor issues:

The article is afflicted with several grammatical and spelling issues, which is unbecoming of a scientific article, specially one that deals with William Shakespeare’s plays. It would be better to go through the article closely and remove any syntactic inaccuracies. A few examples of these are :

in section 1.2, ‘ Ilsemann observed that in these plays the commonest speech length …’

shouldn’t it be ‘most common’ instead of ‘commonest’, as it is much widely used and also avoids ambiguity? In a similar fashion, in the same section, the sentence ‘Lines shared between speakers were rare in Shakespeare’s earliest plays, making up two per cent or so of the total, but became more common in his middle period, and commoner still at the end, with the last plays.’ Makes use of the word ‘commoner’, rather than ‘even more common’. The word ‘commoner’, specially in an article pertaining to Shakespearean plays, could easily be misinterpreted as ‘a common person, as distinguished from one with rank, status, etc.’ and introduce ambiguity in a otherwise lucid and well-articulated article.

In the abstract, the sentence ‘We confirm that that the mode …’ falls prey to one of the more common errors that occur while typing or writing anything in English. The dreaded ‘that that’ in the abstract of an scientific article is disappointing.

I would suggest the authors to comb through the article and rid it of these minor issues before the article is ready for publication.

>We have corrected all the specific errors noted and corrected stylistic errors throughout after a 

>further proof-reading process.

---

## [Editor Report · Decision Letter 2]

22 Feb 2023

Changes in the length of speeches in the plays of William Shakespeare and his contemporaries: a mixed models approach

PONE-D-22-17959R2

Dear Dr. Craig,

We’re pleased to inform you that your manuscript has been judged scientifically suitable for publication and will be formally accepted for publication once it meets all outstanding technical requirements.

Kind regards,

Eugene Demidenko, Ph.D.

Academic Editor

PLOS ONE
---

## [Editor Report · Acceptance letter]

24 Feb 2023

PONE-D-22-17959R2 

Changes in the length of speeches in the plays of William Shakespeare and his contemporaries: a mixed models approach 

Dear Dr. Craig:

I'm pleased to inform you that your manuscript has been deemed suitable for publication in PLOS ONE. Congratulations! Your manuscript is now with our production department. 

Kind regards, 

on behalf of

Dr. Eugene Demidenko 

Academic Editor

PLOS ONE